

# Assessment and quantification of marginal lands for biomass production in Europe using soil quality indicators

Werner Gerwin[1], Frank Repmann[2], Spyridon Galatsidas[3], Despoina Vlachaki[3], Nikos Gounaris[3], Wibke Baumgarten[4], Christiane Volkmann[4], Dimitrios Keramitzis[5], Fotis Kiourtsis[5], Dirk Freese[2]

[1]Research Center Landscape Development and Mining Landscapes, Brandenburg University of Technology Cottbus-Senftenberg, Cottbus, 03046, Germany
[2]Chair of Soil Protection and Recultivation, Brandenburg University of Technology Cottbus-Senftenberg, Cottbus, 03046, Germany
[3]Department of Forestry and Management of the Environment and Natural Resources, Democritus University of Thrace,
Orestiada, 68200, Greece
[4]Fachagentur Nachwachsende Rohstoffe, Gülzow, 18276, Germany
[5]Decentralized Administration of Macedonia & Thrace, Thessaloniki, 54655, Greece

*Correspondence to*: Werner Gerwin (werner.gerwin@b-tu.de)

**Abstract.** The cultivation of bioenergy plants at fertile, arable lands increasingly results in new land use conflicts with food
production and cannot be considered as sustainable, therefore. Marginal lands have been frequently considered as potential alternative for producing bioenergy from biomass. However, clear definitions and assessment methods for selecting marginal lands and for calculating potentials are still widely missing.

The project "SEEMLA" aims at triggering the exploitation of currently underused marginal lands for biomass production for energy purposes. Study sites have been selected in different European countries: Germany, Greece and Ukraine. The selected
sites represent a wide variety of different types of marginal lands. Based on a soil assessment set given by the Muencheberg Soil Quality Rating (SQR) system potentially "marginal" sites have been investigated. The SQR system allows for clearly distinguishing between soils of higher and lower quality. Soils with SQR scores below 40 are regarded as "marginal". They can be classified into different groups with regard to the importance of soil hazard indicators as valuated by the SQR approach. The calculated SQR scores correlate significantly with biomass yields of bioenergy plants.

Further, the SQR method was adapted for use in a GIS study on marginal land potentials in Europe. 46 % of the investigated European area could be classified as "marginal" with SQR scores below 40. From that area 22.6 % can be considered as potentially suitable for producing renewable resources after eliminating protected sites or other places not suitable for any kind of land use. Taking the ecological demands of selected bioenergy plants into account it is possible to give first preliminary recommendations for regional crop cultivation.

It can be concluded, that Europe offers a large potential for renewable resources from marginal sites. However, the implementation into practice is often impeded by missing or varying policies and regulation. A proper implementation needs clear regulations and also incentives for farmers at European level.



## 1. Introduction

With an increasing competition between traditional agriculture for food and feed production and production of renewable resources for bioenergy or biomaterials both unconventional land use systems as well as use of unconventional land gain more and more attention in Europe but also worldwide (Fischer et al., 2009; Popp et al., 2014; Rathmann et al., 2010). As

part of a potential solution for this land use conflict the utilization of lands which are not suitable for conventional high-productivity agriculture increasingly comes into focus. Such lands are frequently addressed as "marginal land" or "surplus land" in recent scientific publications (e.g., Dauber et al., 2012; Kang et al, 2013; Krasuska et al., 2010). However, these terms stand for a large group of very different types of land and definitions for marginal lands are, therefore, very diverse. Several terms are in use, with often synonymous meanings, such as "fallow", "set-aside", "abandoned", "degraded" or

"waste land" (Dauber et al., 2012). Shortall (2013) distinguishes between "land unsuitable for food production", "ambiguous lower quality land" and "economically marginal land". This approach addresses the causes of marginality more explicitly which are also not uniform. They can be the result of land use changes due to technological or socioeconomic transitions (Strijker, 2005) or related with poor natural prerequisites for agriculture. The latter is mainly connected with soil inherent site properties which limit productivity of crop production.

With respect to policies and legal framework, i.e. common agricultural policy (CAP) (EC, 2013b), a definition of marginality of land and its use in agriculture and/or forestry is not given. However, both on member state and EU level, an assessment of land quality and its proper, sustainable, economic and efficient use is needed, especially with respect to feedstock production in marginal or abandoned land, also considering incentive opportunities which have to be tailored for and adapted to the already established policy framework.

Results of assessing potentials of marginal lands suitable for bioenergy production vary widely. As a rough estimation Wolf et al. (2003) stated that about 65 % of the global land area that is potentially suitable for agriculture could contribute to purposes like biomass production. However, this area would be halved if only the presently used arable lands are considered and other functions of land are taken into account, e.g., nature conservation. Particularly, sites which can be classified as "marginal" offer potentials for biodiversity protection and their use might generate new conflicts, e.g., with nature

conservation (Dauber and Miyake, 2016; Miyake et al., 2015; Plieninger et al., 2011).

A careful assessment is needed, therefore, for a proper estimation of potentials of marginal sites for future biomass supply. Different numbers have been reported during the last few years in this regard. Krasuska et al (2010) mentioned a total area available for non-food crops in the European Union of up to 13.2 Mha. According to this study most of this land is allocated in Eastern Europe. For Canada Liu et al (2017) calculated a sum of 9.5 Mha available marginal lands mainly in the southern

part of the country. This latter assumption was based on a soil related approach using the Canadian Land Inventory soil capability classes of the Land Suitability Rating System (LSRS; Agronomic Interpretations Working Group, 1995) as indicators of marginality.



Soil rating systems have been developed in several regions and for different purposes. In Germany, the official land appraisal system ("Bodenschätzung") was developed in the 1930ies to estimate the fiscal value of arable lands and pastures. This methodology is based on a rough estimation of the "average" soil texture of the profile (1 m depth). Data are available nationwide but the transfer of these data to modern soil scientific approaches turns out to be difficult (Hangen and Förster,

2013; Mueller et al., 2010). FAO provides officially the Visual Soil Assessment (VSA) system developed originally by Shepherd (2000). More recently and partly based on the VSA approach the Soil Quality Rating system (SQR) was elaborated by Mueller et al. (2007). This system was developed as a field guide for assessing soil quality with regard to agricultural crop production using up-to-date soil scientific methods. The indicators used in this approach are similar to the bio-physical criteria which were suggested to describe and define natural constraints for agriculture in Europe (van Orshoven et al., 2014)

and which are used by the EU Commission for designating areas facing natural and other specific constraints for agriculture (ANCs; EU regulation No 1305/2013; EC, 2013a). The great advantage of the SQR methodology is that soil quality as a function of a comprehensive set of indicators is expressed in one single number and therefore easily evaluable. For that reason, the SQR system was taken as the basis for an overall estimation and mapping of agricultural yield potentials in Germany carried out by the German Federal Institute for Geosciences and Natural Resources (BGR, 2013).

Mapping of marginal lands has been carried out frequently for different regions and scales as well as with different methodological approaches (e.g., Breuning-Madsen et al., 1990; Cai et al., 2011). In this paper an approach is introduced for assessing and quantifying the area of marginal lands available in Europe for bioenergy production. The presented study is part of the project "Sustainable exploitation of biomass for bioenergy from marginal lands in Europe (SEEMLA)" which aims at triggering the exploitation of currently underused marginal lands for biomass production for energy purposes. One of

the project tasks is the identification of suitable marginal lands as alternative production sites taking into account possible conflicts with competing land use strategies, e.g., food production or nature conservation.

In this regard the objectives of this study are, (i) to identify and test a suitable methodology for discriminating marginal lands, (ii) to characterize and classify marginal lands from a soil-related perspective, and (iii) to apply the methodology for a Europe-wide assessment of marginal land potentials for bioenergy production. For that purpose, the SQR system was applied

and tested to provide soil-related indicators of marginality which can be used for mapping marginal lands in Europe. It was assumed that marginal sites are characterized by low soil fertility or quality as expressed by low or very low SQR scores. It was further assumed, that productivity of such sites is clearly limited for traditional agriculture by means of lacking soil quality so that the mentioned competition between both land use strategies could be mitigated if bioenergy is produced here.

For testing these assumptions soils of marginal sites selected in different European countries were analysed and assessed

according to the SQR methodology. These investigations were thought to confirm the general applicability of the SQR framework for non-agricultural sites and for bioenergy crops, as well. Further, the main limiting factors for the selected marginal land types were evaluated. In a second step a continent-wide mapping of potential areas for bioenergy production



was carried out by means of the SQR methodology adapted according to BGR (2010). The objective was to assess both the overall area of land with low and very low SQR scores and the area of land available without further land use conflicts by excluding protected areas. Finally, recommendations for overcoming further existing policy and legislative constrains are given as an outlook.

## 2. Material and methods

### 2.1 Assessing marginality at case study sites

#### 2.1.1 Case study sites

Case study sites were chosen across three European countries for investigating and demonstrating the practical applicability of bioenergy production on marginal lands. The selection was made to represent different types of marginal lands and different types of climate regimes (Fig. 1 and Table 1). At these sites different methods of bioenergy production were implemented as case studies. At each site soil profiles were analysed, soil samples were taken and soil assessment according to the SQR framework was carried out.

Most of the Ukrainian sites and all investigated Greek sites show limitations for traditional agriculture due to natural constraints and in part due to anthropogenic degradation. The sites in the western part of Ukraine (Volyn and Lviv regions) and in the Poltava region represent the type of abandoned land which was formerly used for conventional agriculture and set aside due to different site specific reasons. In Greece sites with naturally poor soil conditions (e.g., shallow soil depth) were selected in a mountainous region (Rhodope Mountains). The sites are currently in use for forestry or in some parts for low intensity pasture systems.

In contrast, the Vinnitsa site in Ukraine as well as the two German sites has undergone most severe anthropogenic disturbances. The Vinnitsa site was used as municipal waste dump before; waste has been removed before preparing the site. In Lusatia (State of Brandenburg, Eastern Germany) post-mining sites and former industrial or traffic areas represent types of marginal lands which can be frequently found particularly in central and Eastern Europe.

With regard to climatic conditions (cf. Table 1) the selected field sites are within two gradients: (a) between sub-continental and continental and (b) between temperate and Mediterranean types of climate. The Ukrainian sites are characterized by continental climate conditions whereas Lower Lusatia in Eastern Germany has a sub-continentally influenced climate of the temperate zone. Northern Greece is part of the Mediterranean climate region with semiarid climatic conditions.

#### 2.1.2 Soil quality assessment: SQR rating system

To assess soil quality (or vice versa marginality) the Muencheberg Soil Quality Rating system (SQR) (Mueller et al., 2007) was applied using the soil parameters derived from field and laboratory measurements. The SQR is designed to quantify the




soil quality by a single value – theoretically ranging from 1 to 100 points – which is calculated on basis of a set of basic and hazard indicators.

One set of 8 basic indicators describes generic soil parameters (**B 1**: substrate, **B 2**: A horizon depth, **B 3**: topsoil structure, **B 4**: subsoil compaction, **B 5**: rooting depth, **B 6**: profile available water, **B 7**: wetness and ponding, **B 8**: slope and relief).

Single scores are summarized and the resulting basic score has a range between 0 and 34 for arable land. Whereas 0 stands for absolutely infertile soils, 34 can be reached by best suited croplands. In a second step, 13 hazard indicators are examined including further factors influencing soil conditions and ecological functions (**H 1**: contamination, **H 2**: salinization, **H 3**: sodification, **H 4**: acidification, **H 5**: low total nutrient status, **H 6**: soil depth above hard rock, **H 7**: drought, **H 8**: flooding or extreme waterlogging, **H 9**: steep slope, **H 10**: rock at the surface, **H 11**: high percentage of coarse soil texture fragments, **H**

**12**: unsuitable soil thermal regime, **H 13**: miscellaneous hazards). For each hazard indicator the SQR guidelines provide multipliers on a ratio scale between 0.1 and 3. The lowest multiplier found (i.e. the most important hazard for the respective site) is used to calculate the final SQR score which is within a range between 0 and 100. Sites with a final score of 100 can be seen as sites with the best suitable soils for agriculture; whereas soils with SQR score < 40 can be regarded as very poor or poor with regard to agricultural land use (Mueller et al., 2007). Such sites are classified here as "marginal".

**2.1.3 Soil profile description, soil sampling and laboratory analyses**

Soil profiles were investigated at each case study site in summer 2016. Profile descriptions were carried out according to the methodology of the German soil classification system (AG Boden, 2005). The soil horizons' designations were transferred to international symbols according to Blume et al. (2016). Soil types were classified according to international standards provided by the World Reference Base for Soil Resources (WRB) (IUSS Working Group WRB, 2007). Soil samples were

20 taken from dominant soil horizons directly from the profile. From the selected horizons volumetric samples were taken by means of sampling rings for determining bulk density. In addition, mixed samples were taken from different depth of the study sites with different borers (depending from site conditions) and merged. Mixed soil samples were dried (3 days at 40° C) and sieved (< 2 mm). Table 2 gives an overview of parameters assessed by means of field and laboratory methods as needed for soil assessment (SQR). Measured values of relevant parameters are available from Table A1 (Appendix A).

**2.1.4 Soil quality assessment and biomass yield estimation**

For testing the suitability of SQR scores for bioenergy crops mean biomass data were collected for all case study regions. As plantations with bioenergy crops were established at all investigated case study sites later on, direct correlations between field specific SQR scores and biomass yields could not be investigated yet. Thus, as an estimate of local average biomass yields regional project partners provided mean yield results from adjacent field sites with the same bioenergy crops as

cultivated on the respective case study site and with comparable soil conditions. The available biomass yield data were





roughly separated into two groups: biomass yields from woody bioenergy crops (willow, poplar, black locust and pine) and from grass like species (miscanthus, switchgrass). The nonparametric Spearman rank coefficient between SQR final scores and mean biomass yields from woody bioenergy crops (n = 16) was calculated and tested for significance using R statistical package (R Commander). The small data set for grass like bioenergy crops (n = 4) did not allow for calculating correlation

coefficients.

### 2.1.5 Data processing and statistical analysis

SQR score calculations and correlation statistics were performed using MS Excel statistical functions. For deriving typical sets of soil constraints for different types of marginal lands the investigated sites were grouped according to the relevance of their assigned SQR hazard indicators. For this purpose, the assessed SQR hazard multipliers with values between 0.1

(highest influence of this hazard indicator) and 3 (no influence of this hazard indicator) were transformed to a scale of influence (values ranging between 1.0 - hazard of highest relevance for a site, and 0 - no hazard influence detectable). These influence values for the individual hazard indicators were calculated following Eq. (1):

$$Influence = 1 - \frac{x}{3} \qquad (1)$$

with x = value of hazard indicator multipliers.

The resulting transformed values (as summarized in Table A2, Appendix A) were subject to a cluster analysis (Ward method, Squared Euclidian distance) using R statistical package (R Commander). For each cluster mean values of influence were calculated for all hazard indicators.

### 2.2 Concept for quantifying marginal land area in Europe

### 2.2.1 Analysis algorithm

A new algorithm (SEEMLA algorithm) has been developed to characterize marginal lands mainly based on the SQR tool for assessing soil quality. The analysis algorithm investigates land marginality, marginal land availability for biomass production for bioenergy purposes and its suitability for specific bioenergy crops (Fig. 2). Geospatial analysis using Geographical Information Systems (GIS) was applied to identify and display areas of marginal land in Europe.

Marginal land can be clearly defined, based on the scoring scheme of the SQR system. Lands with poor production potential

are considered the ones scoring below 40 in the SQR system. Establishing the availability of marginal land for biomass production however, requires the investigation of certain constraints that can be categorized into three types: physical land constraints (e.g. steep slope that hinders mechanization of works); environmental – ecological constraints (e.g. protected areas); socio-economic constraints (e.g. distance to processing facility, current land use). Such generic criteria have been applied to exclude marginal lands under these constrains and localize areas available for biomass exploitation.

Information regarding the biogeographical region and the biological demands of selected crop species (Table 3) were used to screen the available marginal lands and identify which areas are appropriate for cultivation of the crop species (EUFORGEN, 2016; Korakis, 2015; PFAF, 2016; San-Miguel-Ayanz et al., 2016; Śliż-Szkliniarz, 2013; USDA, 2016). This process cross-references site parameters with the biological demands of the crop species to determine lands suitable for

cultivation of certain bioenergy crops.

The suitability of a marginal land for cultivation of bioenergy crops does not assure sustainable bioenergy production yet. Feedstock quantities and continuous supply, cultivation and harvesting techniques, feedstock transportation and processing to final bioenergy product and other aspects of the whole value chain should be considered. Moreover, the impact of bioenergy production on the environmental, economic and social conditions of a region should be studied. Such concerns

have been included in a set of exploitation scenarios of marginal lands that are currently examined. Each scenario includes life cycle assessment (LCA), life cycle environmental impact assessment (LC-EIA) and an analysis of social and economic aspects, which collectively will determine the sustainability of biomass exploitation.

The SEEMLA algorithm will be further developed in the near future to provide a decision support tool for the exploitation of marginal lands for bioenergy production. This process involves the incorporation of the sustainability assessment results of

the exploitation scenarios in combination with the results from the case study sites. The final step for the algorithm development will be to rate marginal lands, assigning to them priorities for cultivation according to the most efficient exploitation scenario.

### 2.2.2 Geospatial datasets

An extensive search was carried out in order to obtain geographical data for the algorithm components for European

countries. The required geospatial datasets include the basic and hazard indicators for the calculation of the SQR score (Table 2), the screening criteria for the selection of bioenergy crops (Table 3) as well as additional information regarding protected areas and current land cover (Table 4).

Pan-European datasets of the European Soil Data Center (ESDAC) have been primarily used whereas data from the HWSD were used for areas or parameters not covered by the ESDAC datasets, especially for Ukraine. The resolution of the input

datasets varies from 250 m to 5 km.

All input datasets were converted into raster format with the following characteristics:

Cell size: 500 m x 500 m

Spatial reference: ETRS – LAEA 5210          Datum: ETRS89

Latitude of Origin: 52 N          Longitude of origin (Central Meridian): 10 E

Each raster dataset was reclassified based on the SQR field manual, the SQR Assessment scheme according to BGR (2010) and adaptations made by BTU CS within the SEEMLA project.



### 2.2.3 Development of GIS tool

The SEEMLA algorithm was the basis for the development of a GIS toolset to quantify marginal land in Europe and evaluate its potential for biomass production for bioenergy through raster analysis. The necessary geospatial datasets were processed accordingly to allow for implementation of the appropriate SQR calculations and subsequent compilations.

Raster analysis was applied, after the datasets were reclassified in compliance with the SQR classes for basic and hazard indicators. The next step was to apply elimination criteria to derive the suitability of marginal lands for biomass production. The last function of the GIS toolset cross-references the site conditions with crop species demands to determine appropriate marginal lands for cultivating selected crop species.

The outputs of the GIS toolset can be used to produce various thematic maps such as SQR or marginal land map for Europe,
most important hazard indicator, marginal land suitability per bioenergy crop and any desired combination of the resulted layers.

The application was developed as a toolbox for ESRI ArcGIS desktop (v.10.2.2 or newer). The GIS outputs include both raster (cell size 500 x 500 m) and vector datasets and corresponding maps. The standardized outputs include the following information:

1.    Mapping and quantification of marginal lands in Europe using the Muencheberg SQR system

2.    Mapping and quantification of marginal land available for biomass production for bioenergy

3.    Mapping and quantification of marginal land suitable for cultivation of certain bioenergy crops

4.    Mapping and quantification of hazard indicators per raster cell

5.    Mapping and identification of the most important hazard indicator

## 3. Results and discussion

### 3.1 Soils of marginal lands

Soil physical and chemical parameters needed for assessing the SQR indicators are available in Table A1 (Appendix A). In the western Ukrainian regions of Volyn and Lviv sites have been investigated on former arable land which is not further used for agriculture for more than 20 years. The reasons for setting these areas aside are mainly of socioeconomic nature but the
25 studied soils also show clear obstacles for conventional agriculture. Most of the investigated sites in a flat, but some in slightly hilly landscape with an altitude of about 200 m a.s.l. are characterized by sandy substrate and often high groundwater tables. Mires are developed in landscape depressions. With regard to soil types (WRB) both regions are very diverse (Fig. 3) with dominating Cambisols, Regosols and Arenosols as well as Stagnosols. Final SQR scores are within a range from 18.0 to 37.1. The case study site close to Poltava is part of the Chernozem region in the central part of Ukraine.
However, marginal lands can be frequently found in flat hollows with high groundwater tables which impede agricultural





land use. Gleysols developed here with the frequent risk of flooding during spring time. SQR scores of such sites were determined between 55 for better conditions and 1.7 in an extreme situation (regularly completely flooded).

In the mountainous northern part of Greece different very shallow soil profiles have been investigated. In addition, these soils are characterized by very stony texture and are located partly at steep slopes. Most probably the shallow profiles are the

result of erosion processes which occurred previously during periods with a more intensive land use (deforestation, pasture). Today the sites are covered by forests or meadows. The altitude is between 100 and 590 m a.s.l. and the relief is mountainous. Soil types (WRB) found here can be classified as Leptosols or Regosols. The latter is also developed as Colluvic Regosols in small hollows or at the bottom of hillslopes (Fig. 3). Final SQR scores are between 7.6 and 19.3.

Anthropogenic degradation processes were responsible for the formation of soils investigated at the German post-mining and

post-industrial sites. Their soils are characterized by the lack of organic matter and the initially still missing soil structure. Furthermore, they often contain mixed in lignite particles, technogenic material and other coarse substrates like rubble from demolished buildings. At the former waste dump site in the Vinnitsa region of Ukraine several remaining waste particles were still present in the upper part of the soil profile. Frequent soil types (WRB) found in such anthropogenic sites are Regosols or Technosols, both in post-mining landscapes (Spolic Technosols) and in post-industrial sites (Urbic Technosols)

(Fig. 2) and SQR scores vary between 9.1 (post-industrial site) and 29 (Vinnitsa waste dump site).

### 3.2 Assessing marginality

### 3.2.1 Results of SQR soil quality assessment

The SQR assessment of soil quality clearly shows that most of the selected marginal land sites can be in fact considered as "poor" or even "very poor". With only one exception (abandoned arable land in Poltava region, Ukraine: SQR value: 55.0)

all final SQR scores are below the threshold of 40 (Fig. 4). The mean basic score is 17.1 indicating medium soil conditions in general and the mean final SQR score of 22.5 clearly reveals the low quality of soils of marginal lands. For comparison, a mean SQR score of 64 points for arable land in Germany was determined by Hennings et al. (2016). In general, the SQR methodology seems be a generally applicable technique for identifying marginal lands based on soil parameters.

The post-industrial sites (former railway sites, Germany) reached the lowest basic score values as a result of a weak natural

potential of the prevailing artificially created substrates with very young and undeveloped soils. Most of the anthropogenically degraded soils and all mountainous, naturally poor soils (Greece) can be classified as "very poor" with final SQR values below 20, whereas former arable sites usually exhibit considerably better soil quality conditions with SQR scores only slightly below 40. Sites with similar SQR values are frequently found in German low mountain ranges and in the Eastern German low lands (BGR, 2013) at sites which are in use for agriculture. Thus, abandoned arable lands offer at least

some minor potential for traditional agricultural land use options.





The SQR system was primarily invented for traditional agricultural sites and its scores are well correlated with agricultural crop yields (Hennings et al., 2016; Mueller et al., 2010, 2016). By definition marginal lands are not suitable for agriculture and are described as "surplus land" (Dauber et al., 2012). These areas are, therefore, primarily not within the focus of the SQR assessment method. For validating the reliability of the SQR method also for bioenergy crops correlations between

5 (estimated) biomass yields and SQR final scores were tested. Table 5 gives an overview of bioenergy crops cultivated at the studied sites and ranges of the estimated biomass yields as reported by regional project partners for sites with similar degrees of marginality. The estimated local biomass yields of marginal sites used here are in the majority of cases clearly below biomass yields as reported in literature. E.g., Clifton-Brown et al. (2001) found mean biomass yields for miscanthus in Europe of 18 t DM $\cdot$ ha$^{-1}$ $\cdot$ a$^{-1}$. For woody biomass from short rotation plantations in Europe Djomo et al. (2015) published an

10 average of 9.3 t DM $\cdot$ ha$^{-1}$ $\cdot$ a$^{-1}$.

Fig. 5 illustrates the relationship between SQR final scores and estimated biomass yields for the investigated test sites, separately for grass like and woody bioenergy crops. A tendency of growing biomass yields with decreasing marginality (increasing SQR scores) is visible from the figure for grass like bioenergy crops, which are cultivated at two of the Ukrainian test sites. The correlation, however, of woody bioenergy crops with final SQR scores as an indicator of site marginality turns

out to be strong and statistically significant (Fig. 5). It can be preliminary concluded, therefore, that SQR scores are suitable to represent the productivity or vice versa the marginality of soils also with regard to yield potentials of bioenergy crops. Further, it becomes visible from this analysis that poor and also very poor sites with regard to soil conditions provide at least a certain potential for biomass production. The authors are aware of the necessity of an increased number of investigations for being able to derive transferable results and trends.

**3.2.2 Ecological constraints of marginal lands as expressed by SQR hazard indicators**

SQR Hazard indicators are site properties which are able to critically affect the total soil quality (Mueller et al., 2007). Analysing the importance of the different hazard indicators of marginal lands allows for identifying generic factors of marginality which superimpose other properties. Previous studies (BGR, 2013; Hennings et al., 2016) showed that large areas of agricultural land are generally not limited by any hazard indicator. For German agricultural lands Hennings et al.

(2016) calculated an amount of 61.5 % without any ecological constraint according to the SQR assessment. The remaining 38.5 % of arable lands in Germany show different ecological limitations and about 6 % of the arable lands can be seen as clearly marginal with final SQR score below 40. Against this background the relevance of individual hazard indicators for marginality of the selected case study sites presented here was further elaborated. By clustering the investigated case study sites considering their hazard indicators, four groups of marginal lands could be revealed (Fig. 6). The average influence

values of the respective hazard indicators for each cluster are shown in Table 6.



Cluster 1 represents mainly the post-mining sites studied in Lower Lusatia (Germany). Soils at these sites are predominantly affected by higher acidification potentials and a low nutritional status. The same is true for one of the western Ukrainian sites, which is therefore part of this cluster.

The large cluster 2 integrates the further Ukrainian abandoned former arable lands, including also the former municipal
waste dump sites in Vinnitsa region. The main hazards related to these soils are the danger of seasonal over-wetting due to flooding or extreme water logging. Several of these sites are situated in depressions of the flat landscapes and/or exhibit dense layer in the subsurface. The second relevant hazard found here was the insufficient nutritional status of the soils. Both factors can be seen as the reason for abandoning these former agricultural sites during the last two decades.

Extreme soil conditions were found at the German post-industrial site (former railway area), which was grouped into cluster
3. Main hazards here are the very high amount of coarse particles above and in the soil substrate. These particles consist of remains of railroad ballast as well as of rubble from former buildings at this site. This composition can be seen as typical for brownfields. The investigated site showed no signs of contamination.

Cluster 4 contains the Greek sites, which were already characterized as marginal sites with naturally poor soil conditions. With its Mediterranean climate regime Greece is generally prone to droughts during summer season. Furthermore, soil
substrates are in part extremely stony both at the surface and within the profile. It is assumed that the geology of the Rhodope Mountains in northern Greece with frequent ore deposits provides slightly elevated background contents of heavy metals (Cr, Ni) and arsenic (As) so that the contamination hazard indicator gained higher influence. In addition, as a result of previous erosion the remaining soil profiles are mainly shallow with low depths above hard rock. This is consistent with conclusions by Hennings et al. (2016) who found low soil depth as important SQR hazard indicator in mountain regions of
Germany.

### 3.3 Quantification of marginal land potentials in Europe

Marginal land in Europe was quantified using the developed GIS tool. The first outcome was the calculation of the SQR index incorporating all 8 basic indicators and 11 hazard indicators (H 2: salinization, H 3: sodification, H 4: acidification, H 6: soil depth above hard rock, H 7: drought, H 8: flooding or extreme waterlogging, H 9: steep slope, H 10: rock at the
surface, H 11: high percentage of coarse soil texture fragments, H 12: unsuitable soil thermal regime and H 13: disturbance by man). The produced map with the SQR value classes is presented in Fig. 7. Approximately 257 Mha of land in Europe[1] belongs to the poor and very poor classes of the SQR index and is identified as marginal. This area corresponds to 46 % of the overall area investigated.

---

[1] Albania, Andorra, Austria, Belgium, Bosnia Herzegovina, Bulgaria, Croatia, Czech Republic, Denmark, Estonia, Finland, France, FYROM, Germany, Greece, Hungary, Ireland, Italy, Latvia, Liechtenstein, Lithuania, Luxembourg, Malta, Monaco, Montenegro, Netherlands, Norway, Poland, Portugal, Romania, San Marino, Serbia, Slovakia, Slovenia, Spain, Sweden, Switzerland, United Kingdom, Ukraine



The marginal land available for biomass production for bioenergy purposes, calculated after implementation of the proper elimination criteria (protected areas, specific land uses and slope steepness), is depicted in Fig. 8. The area of the sites with marginal soil conditions available for biomass production is approximately 58.2 Mha. This figure represents 10.5 % of the overall area of Europe and 22.6 % of the overall marginal land in the region.

The countries with the greatest ratio of marginal lands in relation to their overall area (>20 %) are in descending order: San Marino, Albania, Portugal, Italy, Lithuania and Norway (Fig. 9). The corresponding percentages for the countries with SEEMLA case study sites are 14.6 % (1.9 Mha) for Greece and 9.4 % (3.3 Mha) for Germany, while for Ukraine these figures could not be calculated due to the lack of geospatial data regarding the availability of marginal land for biomass production in the country.

According to the SQR methodology, land marginality can be attributed either to unfavourable soil substrates or to the influence of specific hazard indicators. The GIS analysis showed that poor soil conditions, not influenced by any hazard indicator of the SQR system, account for 34 % (19.7 Mha) of marginal lands available for biomass production in Europe, whereas areas with marginality owed to one or more hazard indicators account for the remaining 66 % (38.6 Mha) (Fig. 10). Generic soil parameters are jointly considered in the calculation of the SQR value as the weighted sum of the basic

indicators, with values below 13 to indicate land marginality (1.03 Mha). In case of missing values for any of the basic indicators the SQR score is not calculated, ensuring that all basic indicators are taken into account.

Hazard indicators on the other hand have an additive effect to the SQR calculation, in the sense that a single hazard indicator is sufficient to classify an area as marginal. Thus, applying as many of these potential marginality factors as possible provides a more precise estimation. Even though the overall area affected by a specific hazard may be limited, its importance

can be significant at local level. The importance of each hazard indicator should therefore be assessed locally, regardless of the overall area it affects, based on its scale of influence (values ranging between 1.0 - hazard of relevance for a site, and 0 - no hazard influence detectable). Hazard indicators with scale of influence over 0.5 account for land marginality in most European countries (Fig. 10).

The most frequent hazard indicator identified in Europe was H 6: Soil depth above hard rock, followed by H 12: unsuitable

soil thermal regime and H 7: drought risk (Table 7 & Fig.11).

The overall area of influence of the hazard indicators is slightly higher due to the fact that in some sites there are more than one determinant marginality factors, accounting for the difference of 0.25 Mha arising between the two numbers reported (38.6 Mha overall area versus 38.85 Mha in Table7).

While the impact of H 6 (Soil depth above hard rock) extends to all of Europe, the influence of the other two hazard

indicators is localized. Drought risk (H 7) is a significant marginality factor for the Mediterranean (mostly Spain and Italy), whereas unsuitable soil thermal regime accounts for the marginality of most areas in Nordic countries (mostly Finland and Sweden) (Fig. 11).



The soil substrates, in combination with the marginality factors and ecological demands of the plants, determine the bioenergy crops that are suitable for each marginal land. The criteria that were taken into account for the selection of the bioenergy crops include: drought risk, pH, wetness and ponding, altitudinal range and biogeographical region. The threshold values for the selected bioenergy crops are cross-referenced with the site parameters in order to locate the areas of interest.

An overview of the marginal land that is suitable for selected bioenergy crops (basket willow, black locust, black pine, miscanthus, poplar and switchgrass) is presented in Fig.12. The area has been classified per species or species group in order to map those with the wider range in Europe, taking into account that in most cases more than one plant species can be grown in a specific site.

The area of marginal land that can be exploited for biomass production with the use of the selected bioenergy crops is

10 approximately 44.62 Mha, which is less than 80 % of the overall marginal land that was originally identified as suitable (58.2 Mha). The ecological demands of the plant species act as another constraining factor for marginal land exploitation, further reducing the available area. Based on results of the GIS analysis, poplar can be grown in most of the sites, contrary to black pine and miscanthus (Table 8). Land marginality is a fundamental limiting factor for plant growth. Therefore, the selection of the most adapted species to these extreme conditions is of major importance to sustainable biomass production.

**4. Outlook on policies and legal frameworks for utilization of marginal land**

In the previous sections the potential availability of marginal land for biomass production has been illustrated. However, current EU and national policies neither include nor describe explicitly aspects of marginal land use. Instruments for mobilising and fostering the use of marginal in terms of abandoned, degraded, economically inefficient land are missing. Nevertheless, the overall framework for a sustainable biomass production for bioenergy is already established in the EU's

Renewable Energy Directives (RED; EC, 2009) and the Member States' National Renewable Energy Action Plans (NREAP; EC, 2010), as well as in national legislation regarding environmental protection (incl. soil, water, air/atmosphere), biodiversity, agriculture, and forestry. However, an increasing demand for biomass needs to be related to an individual biomass potential in each country, respectively. So far, the estimated potentials vary a lot between European Member States, and individual strategies are required to bridge the gaps of future energy demand (Scarlat et al., 2015). Therefore, the

exploitation of biomass production for bioenergy in marginal land offers a great potential in order to be able to meet the 2020 targets and beyond.

With regard to biomass production in marginal lands, an early stage of market deployment has to be considered. In Table 9 corresponding policy mechanisms related to the use of marginal land for bioenergy purposes are given. At this stage of initial market development, a strategy and action plan are crucial. Main financial support tools are R&D grants, investment

subsidies, loans or credit lines, followed by tax exemptions in the transgression state from the initial to the early stage.

However, major aims are (i) to create an attractive incentive programme for stakeholders, i.e. farmers, foresters, to use marginal lands for sustainable biomass production for bioenergy, (ii) to share experiences in each country internally and transnationally with other EU member states, and to (iii) apply the EU Common Agricultural Policy (CAP; EC 2013b), finding a way to adapt "marginality" to the CAP, e.g. "greening"', and other relevant legislation, e.g. European water

protection, nature conservation, soil protection, nitrates directive and related regulations, frameworks, and financial supporting programs.

## 4.1 Regulations suitable for biomass from marginal lands

Based on recommendations of the BiomassPolicies project[2] and adapted to SEEMLA, it has to be ensured that CAP measures from pillar 1 "direct payments" and pillar 2 "rural development" are integrated into local planning and that there

are provisions for biomass from marginal lands. In detail, regarding CAP pillar 2 targeted national and/or regional rural development programs should be introduced (where they are not existing) focusing on shift to low-carbon economy, including targeted measures for municipalities. Action plans are to be developed including all measures dealing with the use, management, conservation and protection of planted public areas where biomass production on marginal lands exists. Biomass certification activities need to be enhanced at national level, whereas national preconditions are better taken into

account by national policy.

## 4.2 Financial support measures suitable for biomass from marginal lands

With regard to CAP pillar 1 for direct payments, budget from "Green Direct Payments" should include appropriate marginal land management activities matched to regions, municipalities, local ecosystems and practices. This will lead to optimized biomass mobilization. Subsidies for improving the types of species in marginal lands with [indigenous] fast growing species

should be offered as well as for SMEs, cities, municipalities etc. in order to purchase equipment for improved harvesting and handling (chippers, pelletizers, etc.) operations. The preparation of a marginal land management plan need to be supported and be provided in form of grant or tax exemptions for improving existing biomass trade centres to include such biomass forms in their selling products, etc.

## 4.3 Information provision mechanisms suitable for biomass from marginal lands

Knowledge should be transferred and human resource capital at local, municipality level should be improved. Capacity building for improved practices should be provided with regard to quality, handling and storage of biomass from marginal lands, as well as capacity building to existing wood trade centres on handling biomass from marginal land. Learning from

[2] see: http://www.biomasspolicies.eu/tool/about_project.html



good practices is of great relevance for a successful development of a suitable SEEMLA approach and the foreseen introduction to the policy framework of the partner countries and, in mid-term perspectives, to EU legislation.

## 4. Conclusions

The study illustrates the large potentials but also some constraints of European marginal lands for contributing to a future energy supply based on renewable resources. In addition, the importance of a proper definition of marginal lands to be used as alternative areas for biomass production becomes clear, too. Sites to be announced as "marginal" vary extremely with regard to their soil properties and land use potentials. A number of potential soil related hazards were identified to be the reason of marginality and also to limit the use of such sites for bioenergy production. In this regard the SQR assessment method demonstrated clearly its ability to differentiate between fertile lands suitable for traditional agriculture and food production on the one hand and marginal sites which offer opportunities for future bioenergy production on the other hand. Based on these results potentially suitable areas for selected bioenergy crops were identified by means of low SQR scores using an innovative GIS approach. Generally, the SQR method turns out to be easy to apply during local field work as well as being useful for GIS assessment studies on site potentials at continental scale. A clear advantage of the SQR system is its easy adaptability for GIS studies. The choice of input data, however, is still a demanding issue as soil related data sets with higher spatial resolution are not available for all European countries. Further development of this GIS assessment approach will also need careful ground truthing of its results. Several open questions remain particularly with regard to the accuracy of the input data and the resulting local soil classification.

Furthermore, open research questions remain with regard to the correlation between biomass yields and SQR scores. As the SQR method was primarily developed for agricultural crops, adaptations of the assessment system might be necessary with regard to the ecological demands of bioenergy crops. Generally, yield data presented here suggest that sites with low SQR scores have a clearly reduced productivity also with regard to biomass yields of bioenergy crops. Therefore, the efficiency of biomass production must be considered as low at sites with very unfertile soils and might appear not very attractive for farmers to invest. However, examples from Eastern European countries illustrate that the "re-cultivation" of set-aside lands with medium marginality might be an attractive option for local and regional value chains and socioeconomic systems.

It is important to note that new conflicts might arise if soils, which are unproductive for agriculture under present socioeconomic terms, are thought to be utilized for future bioenergy crop production. Often, marginal lands have been set-aside already for a long period of time or have been excluded from agriculture traditionally so that rare species and seldom habitats are frequently found. Such sites are usually subject to nature conservation measures. A proper selection of suitable marginal lands and clear legal regulations are therefore crucial and explicitly needed for a sustainable production of energy from renewable resources. However, the proposed alternative land use systems for suitable marginal lands are supposed to

be low input systems and can be considered to have, therefore, positive impacts on soil protection compared to intensive conventional agricultural systems (Fernando et al., 2018).

The practical implementation of a sustainable biomass production for energetic use at marginal lands is, therefore, highly depending on economic trade-offs as well as on supportive policy mechanisms and the framework of regulations. At present,

a wide range of different national policies and regulations exists within European countries. This impedes an overall implementation of this approach and causes unnecessary reservations of local farmers, landowners and authorities. Furthermore, additional financial support and incentives for farmers are still widely missing to overcome the relatively low efficiency of marginal soils compared with highly productive – but for food production needed – soils. Integrating bioenergy production at suitable marginal lands into future European policies (CAP) and the creation of suitable incentive programs

might contribute to the objective to reach national and European renewable energy goals and to mitigate the rising land use conflict between the production of food and feed on the one hand and biomass on the other hand. It can be expected that the importance of marginal lands will increase during the next few decades as bioenergy is thought to play an important role for future energy supply in Europe.

## Author contribution

Field work, soil analysis and quality assessment at case study sites was carried out by WG and FR, locally assisted by FK and DK. The GIS study was performed by SG, DV and NG. The legal and policy framework was elaborated by WB with support by CV. DF contributed to data analysis and interpretation.

## Competing interests

The authors declare that they have no conflict of interest.

## Disclaimer

This paper reflects only the authors' view and the Innovation and Networks Executive Agency (INEA) as delegated by the European Commission is not responsible for any use that may be made of the information it contains.





**Acknowledgements**

This project "SEEMLA" received funding from the European Union's Horizon 2020 research and innovation programme
under grant No 691874. The authors would like to thank all SEEMLA project partners for their support during field work
and sampling as well as for providing biomass yield data for the investigated case study sites.

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



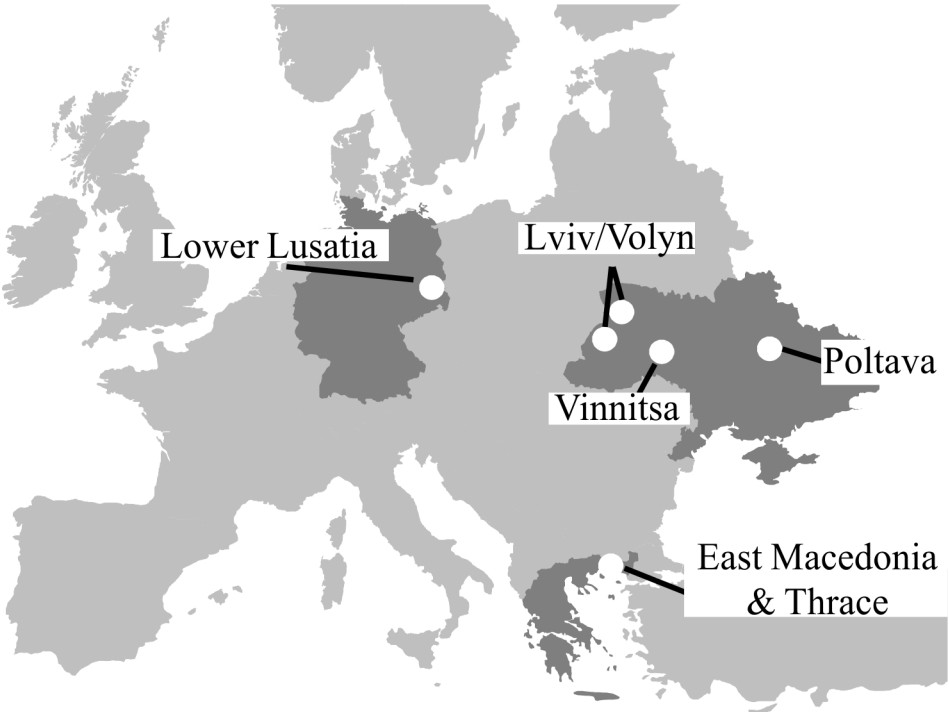

**Figure 1: European regions with investigated case study sites.**




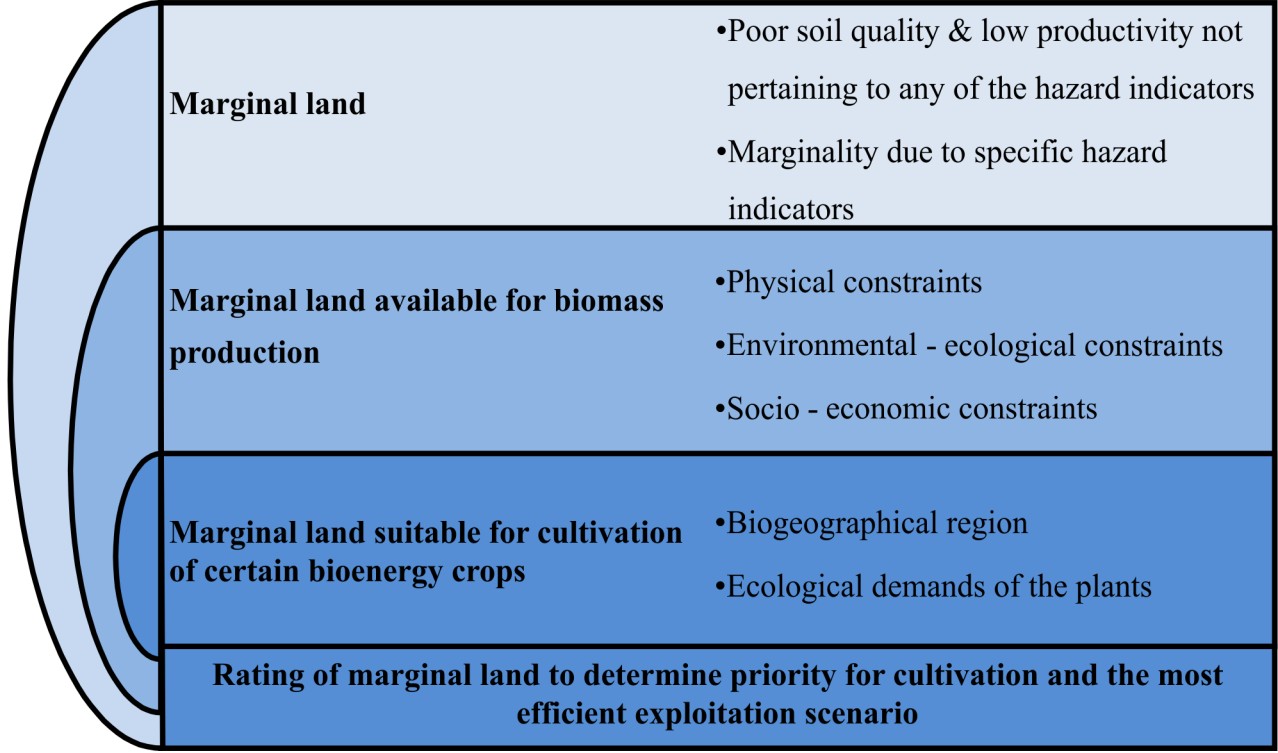

**Figure 2: SEEMLA algorithm.**



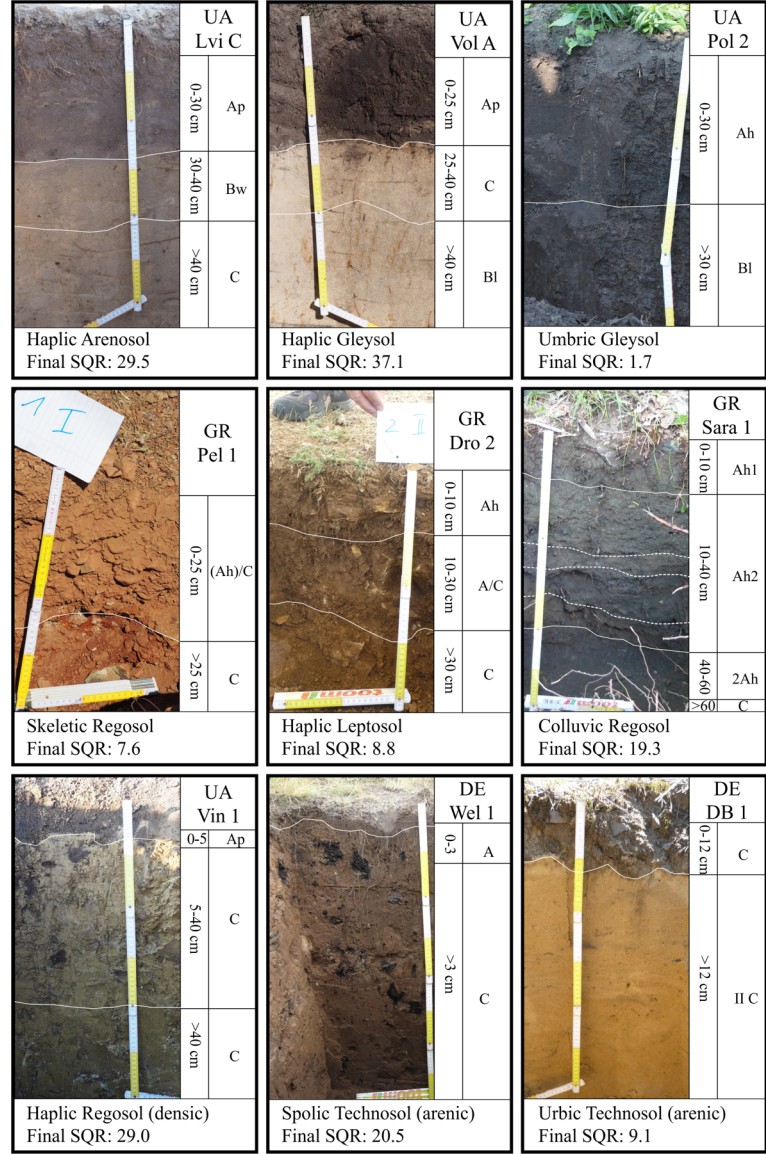

**Figure 3: Soil profiles of marginal lands: above: abandoned arable lands in Ukraine (UA); middle: mountain soil profiles in Greece (GR); bottom: anthropogenically degraded soils in Ukraine (left) and Germany (DE).**



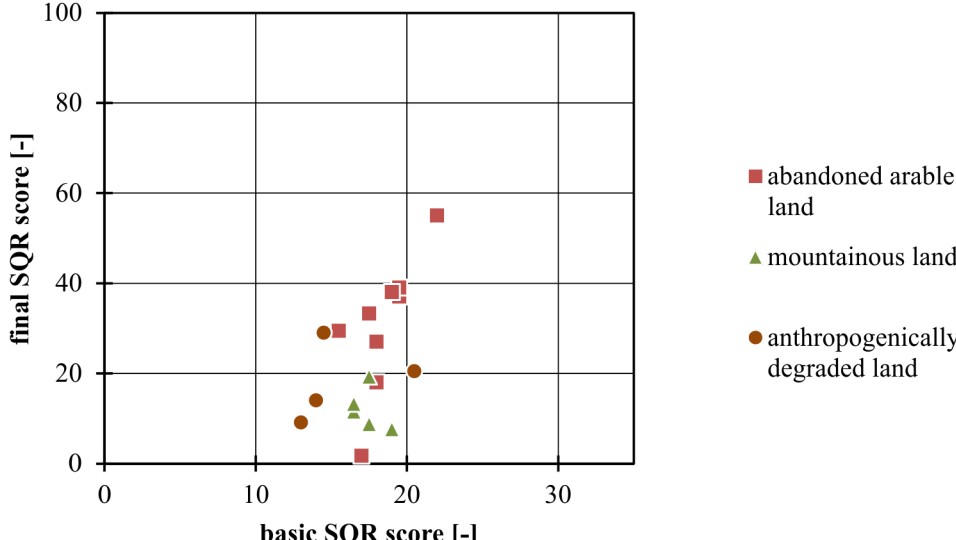

**Figure 4: Overview of SQR basic and final scores for investigated marginal case study sites.**





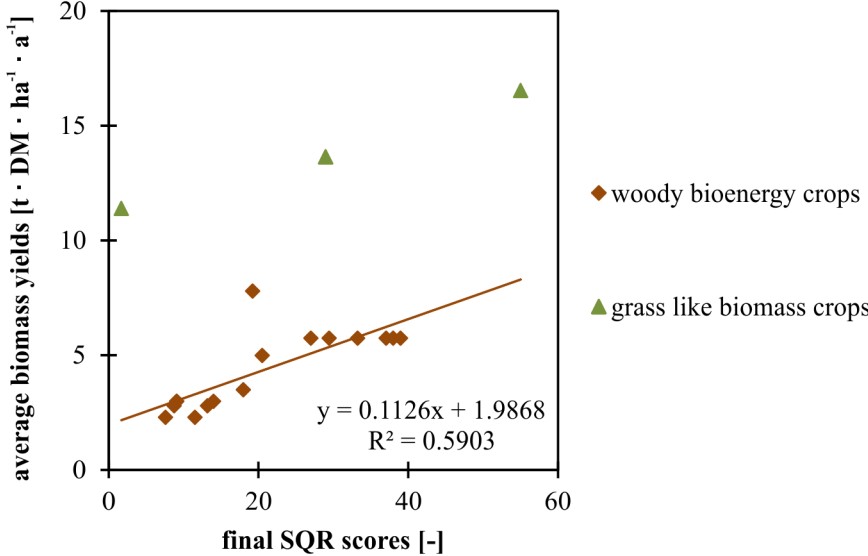

**Figure 5: Correlation between final SQR scores and estimated average biomass yields for woody bioenergy crops with linear regression line (n = 16, r = 0.84\*\*, p < 0.01) and grass like bioenergy crops (n = 4, no correlation coefficient available.).**




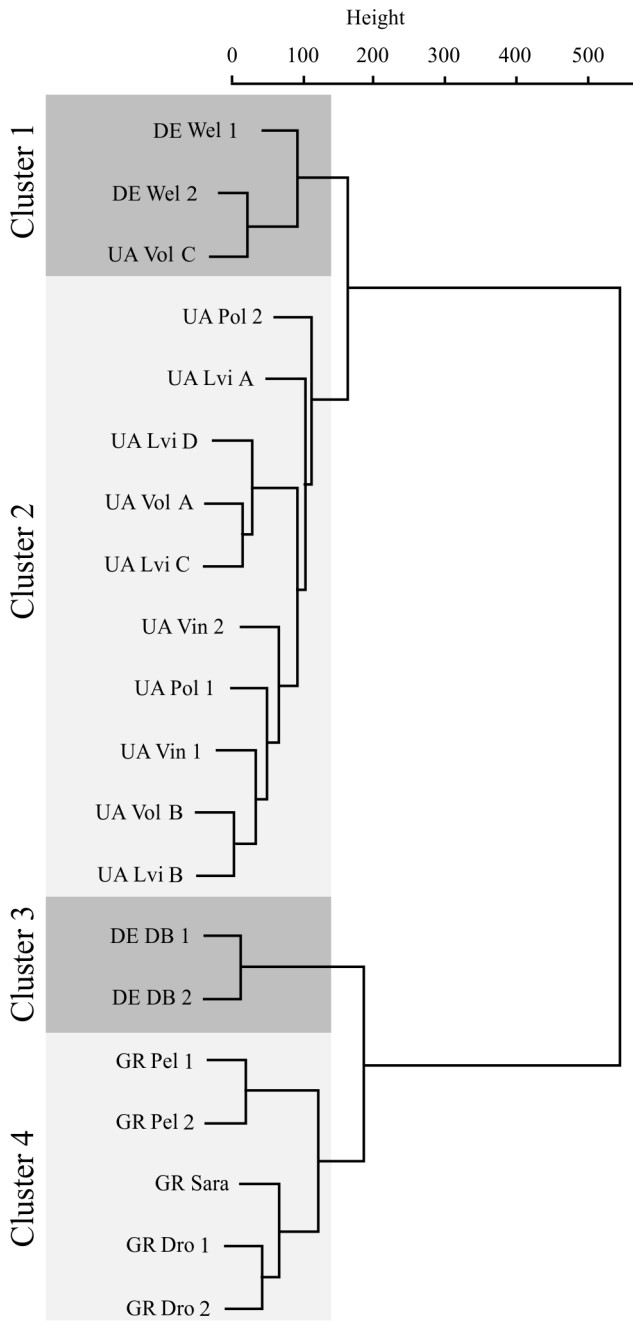

**Figure 6: Investigated marginal case study sites grouped into four clusters based on influence values of their SQR hazard indicators.**



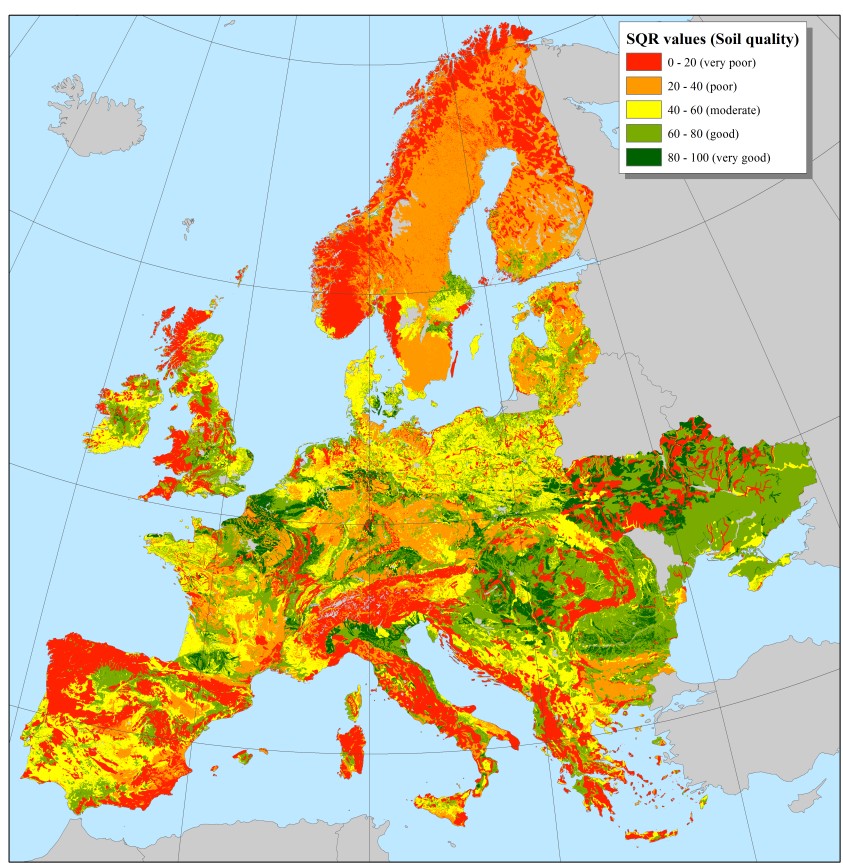

**Figure 7: SQR calculation results for Europe taking into account all basic and 11 hazard indicators**





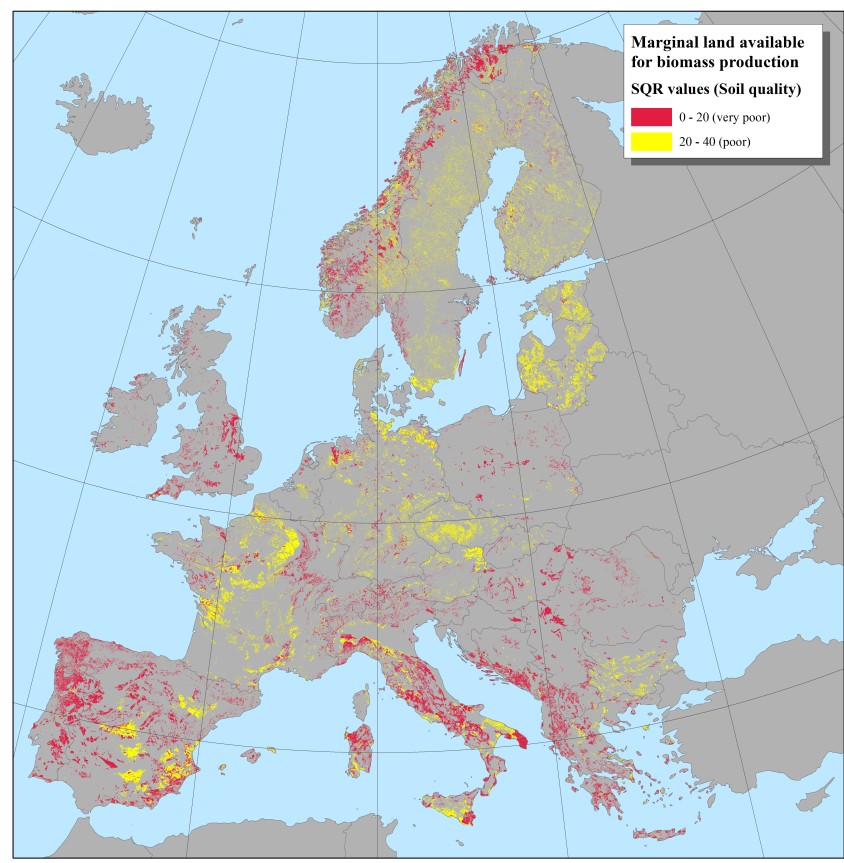

**Figure 8: Marginal land available for biomass production for bioenergy purposes in Europe**



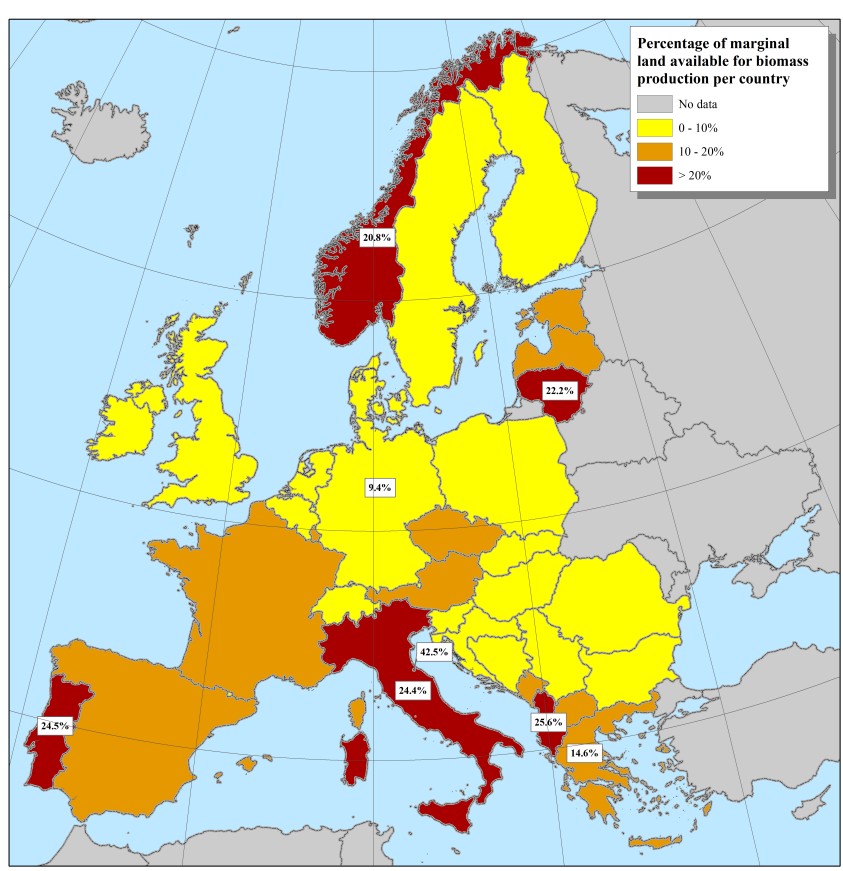

**Figure 9: Percentage of marginal land available for biomass exploitation per country**




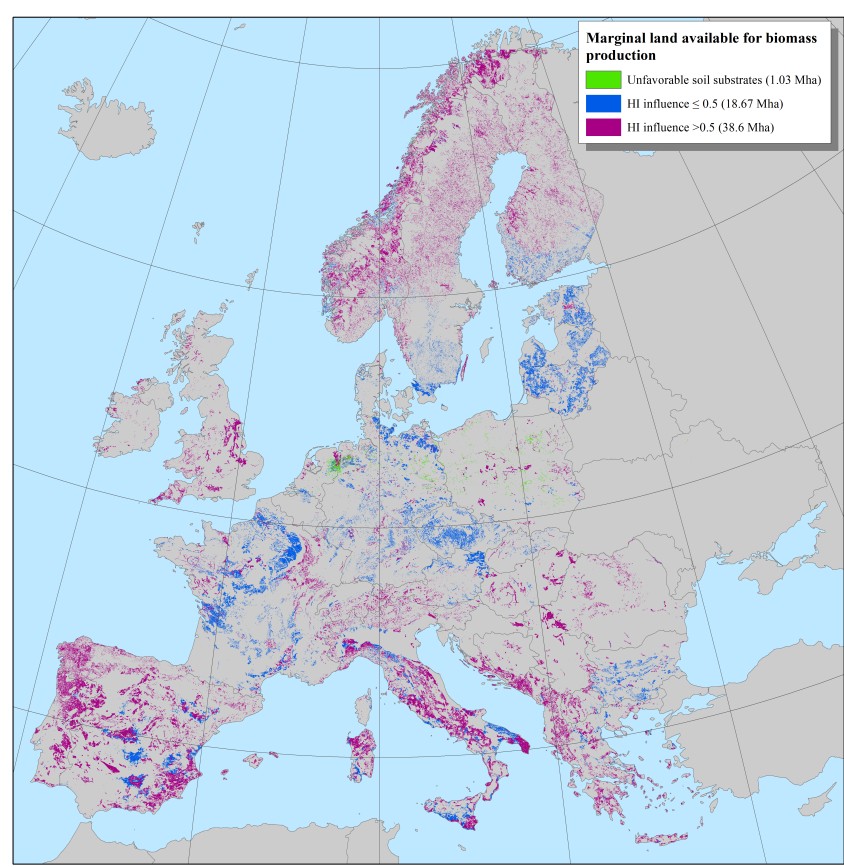

**Figure 10: SQR hazard indicators area of influence as marginality factors in Europe**



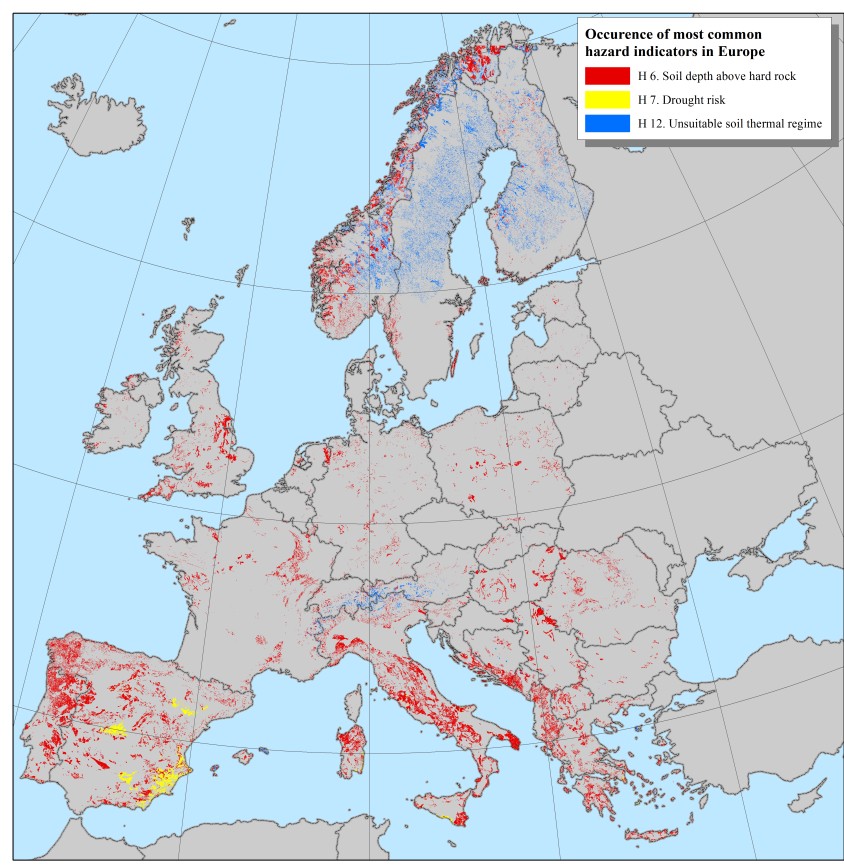

**Figure 11: Occurrence of the most common hazard indicators (H 6, H 7 & H 12) in Europe**



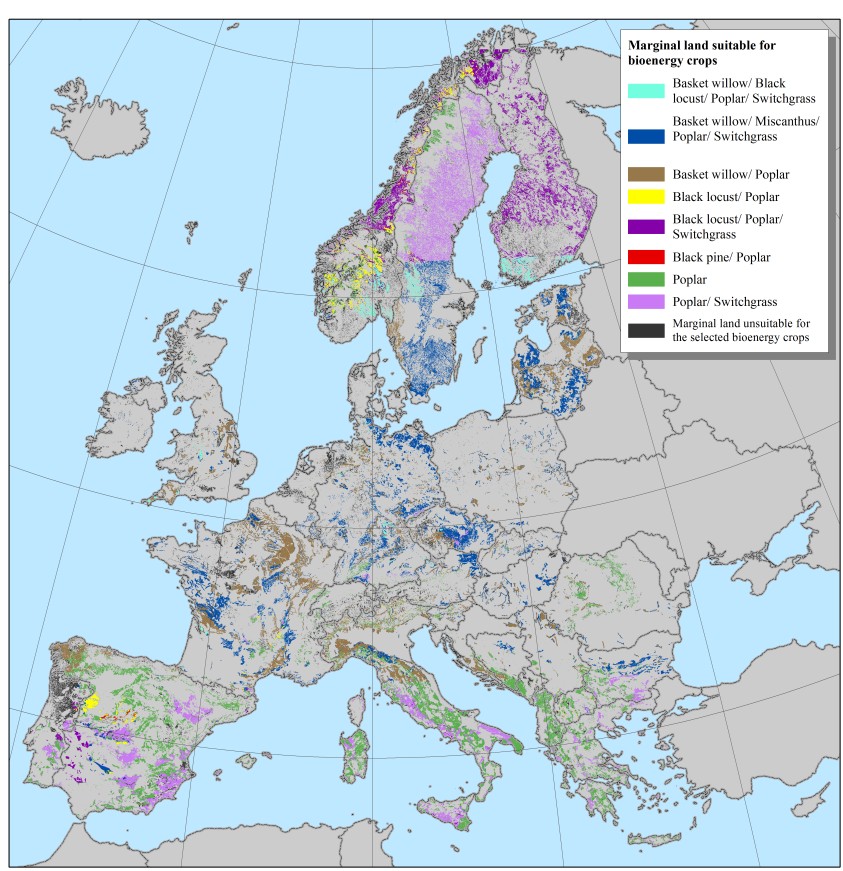

**Figure 12: Marginal land suitability for bioenergy crops**





**Table 1: Overview of investigated case study sites: Locations, climatic conditions and cultivated bioenergy crops.**

| Region | Local name (village/ town) | Site and profile numbers | Mean annual temperature / precipitation | De Martonne Aridity Index | Type of bioenergy production planned |
|---|---|---|---|---|---|
| **Ukraine** | | | | | |
| Vinnitsa | Yaltushky | UA Vin 1 UA Vin 2 | 6.9° C / 529 mm | 28.5 | Perennial bioenergy crops (*Miscanthus × giganteus*, *Panicum virgantum*) / energy trees (*Salix sp.*) |
| Poltava | Semeniwka | UA Pol 1 UA Pol 2 | 7.7° C / 511 mm | 33.0 | |
| Volyn | Zubylne/ Kysylyn | UA Vol A UA Vol B UA Vol C | 9.5° C / 610 mm | 31.1 | Energy trees (*Salix spec.*, *Populus sp.*) |
| Lviv | Welyki Mosty | UA Lvi A UA Lvi B UA Lvi C UA Lvi D | 6.9° C / 668 mm | 37.7 | |
| **Greece** | | | | | |
| East Macedonia & Thrace | Sarakini Drosia Pelagia | GR Sara 1 GR Dro 1 GR Dro 2 GR Pel 1 GR Pel 2 | 14.8° C / 672 mm | 16.8 | Energy trees (*Pinus sp.*, *Robinia pseudoacacia*) |
| **Germany** | | | | | |
| Lower Lusatia (State of Brandenburg) | Welzow Cottbus | DE Wel 1 DE Wel 2 DE DB 1 DE DB 2 | 8.9° C / 563 mm | 32.1 | Energy trees (*Robinia pseudoacacia, Populus sp.*) |

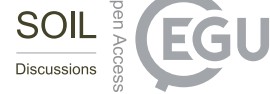



**Table 2: Soil parameters assessed in the field and as basis for assessing SQR indicators.**

| Parameter | Methodology | SQR indicators |
|---|---|---|
| Soil texture | Particle size analysis after destruction of carbonates (application of HCl) and organic substances (application of $H_2O_2$), suspension with $Na_4P_2O_7$, determination of sand fractions by sieving (< 2-0.63 mm, < 0.63-0.2 mm and < 0.2-0.063 mm, determination of silt and clay by sedimentation analysis according to Köhn (Blume et al., 2011). | B 1 |
| Soil depth | Measurement (soil profile: A horizon, depth of hard rock) | B 2, H 6 |
| Soil structure | Visual assessment according to Lists 19-22 of German Soil Mapping Guidelines (AG Boden, 2005) | B 3 |
| Bulk density | Weighing volumetric soil samples (volume of sampling rings: 100 cm³) after drying (Blume et al., 2011), converted into packing density values according to Harrach et al. (1999) | B 4 |
| Rooting depth | Effective rooting depth assessed according to Table 81 of German Soil Mapping Guidelines (AG Boden, 2005) | B 5 |
| Profile available water | Assessment of plant available field capacity according to Table 70 of German Soil Mapping Guidelines (AG Boden, 2005), water volume calculated for the effective rooting depth | B 6 |
| Hydromorphic features | Visual inspection of profile and soil colours (occurrence of mottles, rusty or pale colours, dark concretions etc.) | B 7 |
| Relief situation | Assessment of slope inclination according to Table 6 of German Soil Mapping Guidelines (AG Boden, 2005) | B 8, H 9 |
| Element content (heavy metals) | Measurement in solid soil samples using x-ray-fluorescence device (NITON XL3t analyser, Thermo Fisher) | H 1 |
| $EC_{2.5}$ | Measurement in 1 : 2.5 soil : water suspension (Blume et al., 2011), conversion into $EC_{SE}$ values according to FAO (2006) | H 2 |
| $pH_{H2O}$ | Measurement in 1 : 2.5 soil : water suspension (Blume et al., 2011). | H 3, H 4 |
| Plant available nutrients (P, K) | Measurement in solutions from lactate extraction (Blume et al., 2011) using ICP OES | H 5 |
| Coarse fragments | Sieving of substrate (> 2 mm) from the surface and the soil profile | H 10, H 11 |
| Climate | Analysis of climate data provided for each site: calculation of De Martonne Aridity Index (Mueller et al., 2007) and of duration of frost free period | H 7, H 12 |
| Ponding | Visual inspection of the site: indications of waterlogging at the surface | H 8 |



**Table 3: Selected bioenergy crops screening criteria – specific growing conditions.**

| Common name | Scientific name | Climatic zone | Optimal altitude (min - max) [m a.s.l.] | Optimal soil pH (min - max) | Optimal soil moisture conditions |
|---|---|---|---|---|---|
| Black locust | *Robinia pseudoacacia* | Atlantic Continental Mediterranean | 0 - 1040 m | 4.5 - 8.2 | well-drained |
| Black pine | *Pinus nigra* | Atlantic Continental Mediterranean | 350 - 2200 m | 4 - 8 | dry |
| Basket willow | *Salix viminalis* | Atlantic Continental | 0 - 570 m | 5 - 7.5 | well-drained to wet |
| Poplar | *Populus sp.* | Atlantic Continental Mediterranean | 0 - 1200 m | 4.5 - 7.5 | wet |
| Miscanthus | *Miscanthus × giganteus* | Atlantic Continental | 0 - 1000 m | 5.5 - 7.5 | moist, well-drained |
| Switchgrass | *Panicum virgatum* | Atlantic Continental Mediterranean | 0 - 700 m | 5 - 7 | moderately to well-drained |



**Table 4: Overview of geospatial datasets used and corresponding sources needed for SQR basic and hazard indicator valuation (B 1-8; H 1-13).**

| Datasets | Extent | Data source |
|---|---|---|
| Soil properties (B 1- B7, H 1 - H 6, H 9, H 11 & bioenergy crop demands) | European<br>Global | ESDAC European Soil Database distribution v2.0<br>FAO Harmonized World Soil Database (HWSD) v 1.2 |
| Limitations to agricultural use (H 8, H 10 & H 13) | European | ESDAC European Soil Database distribution v2.0 |
| Climate data (H 7) | Global | WorldClim - Global Climate Data |
| Köppen-Geiger Climate Classification (H 12 & bioenergy crop demands) | Global | Institute for Veterinary Public Health |
| Slope (B 8, H 9) | Global | NASA-Shuttle Radar Topography Mission (SRTM) digital elevation model |
| Protected areas in Europe | European | European Environment Agency (EEA) |
| Corine Land Cover v.18.5.1 | European | EEA Copernicus programme |





**Table 5: Average biomass yields at case study sites: Ranges for different degrees of soil marginality (data provided by regional project partners).**

| Region | Local site name (village/ town) | Cultivated bioenergy crop | Biomass yields [t DM · ha⁻¹ · a⁻¹] |
|---|---|---|---|
| | | **Ukraine** | |
| Poltava | Semeniwka | *Panicum virgatum* / *Miscanthus × giganteus* | 10.0 … 18.1 |
| Vinnitsa | Yaltushky | *Panicum virgatum* / *Miscanthus × giganteus* | 12.0 … 15.3 |
| Volyn | Zubylne/ Kysylyn | *Salix* sp. / *Populus* sp. | 5.5 … 6.0 |
| Lviv | Welyki Mosty | *Salix* sp. / *Populus* sp. | 3.5 … 6.0 |
| | | **Greece** | |
| East Macedonia & Thrace | Drosia | *Pinus nigra* | 7.2 … 8.3 |
| | Pelagia | *Pinus brutia* | 2.8 |
| | Sarakini | *Robinia pseudoacacia* | 2.3 |
| | | **Germany** | |
| Lower Lusatia (State of Brandenburg) | Welzow | *Robinia pseudoacacia* | 3.0 … 5.0 |
| | Cottbus | *Robinia pseudoacacia* / *Populus* sp. | 3.0 |





**Table 6: Mean influence values (0: no influence; 1.0: maximum influence) of hazard indicators (H 1 -11) for clusters of marginal case study sites (numbers in bold are the dominating values for the respective cluster).**

| | H 1 | H 2 | H 3 | H 4 | H 5 | H 6 | H 7 | H 8 | H 9 | H 10 | H 11 |
|---|---|---|---|---|---|---|---|---|---|---|---|
| | Contamination | Salinization | Sodification | Acidification | Soil nutrient status | Depth above rock | Drought risk | Flooding/ waterlogging | Slope | Stones at the surface | Coarse soil fragments |
| **Cluster 1** | 0.00 | **0.22** | 0.00 | **0.49** | **0.23** | 0.00 | 0.00 | 0.01 | 0.00 | 0.00 | 0.00 |
| **Cluster 2** | 0.04 | 0.04 | 0.02 | 0.00 | **0.13** | 0.07 | 0.01 | **0.28** | 0.00 | 0.03 | 0.03 |
| **Cluster 3** | 0.03 | 0.05 | 0.00 | 0.00 | 0.05 | 0.00 | 0.00 | 0.00 | 0.00 | **0.50** | **0.77** |
| **Cluster 4** | **0.45** | 0.00 | 0.00 | 0.00 | **0.20** | **0.34** | **0.63** | 0.00 | 0.07 | **0.39** | **0.71** |

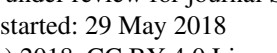



**Table 7: Extent of different soil related hazard indicators for European marginal lands.**

| Hazard indicator | | Area [Mha] | Marginal land influenced at European scale [%] |
|---|---|---|---|
| H 2 | Salinization | 0.05 | 0.09 % |
| H 3 | Sodification | 0.04 | 0.07 % |
| H 4 | Acidification | 0.19 | 0.32 % |
| **H 6** | **Soil depth above hard rock** | **27.86** | **47.34 %** |
| **H 7** | **Drought risk** | **1.87** | **3.19 %** |
| H 8 | Flooding or extreme waterlogging | 0.25 | 0.42 % |
| H 9 | Steep slope | 0.08 | 0.13 % |
| H 10 | Rock at the surface | 0.26 | 0.44 % |
| H 11 | High percentage of coarse soil texture fragments | 0.07 | 0.13 % |
| **H 12** | **Unsuitable soil thermal regime** | **8.12** | **13.80 %** |
| H 13 | Disturbance by man | 0.04 | 0.07 % |
| | Sum | 38.85 | 66,00 % |





**Table 8: Marginal lands available for biomass production per bioenergy crops.**

| Bioenergy crop | Area [Mha] | Percentage of marginal land available for biomass production [%] |
|---|---|---|
| *Salix viminalis* | 18.57 | 41.88 % |
| *Robinia pseudoacacia* | 6.52 | 14.70 % |
| *Pinus nigra* | 1.28 | 2.87 % |
| *Miscanthus × giganteus* | 5.36 | 12.01 % |
| *Populus sp.* | 43.15 | 97.32 % |
| *Panicum virgatum* | 21.05 | 47.47 % |



**Table 9: Policy mechanisms relevant to biomass from marginal land per value chain step, type of policy and market stage development (adapted from BiomassPolicies.eu, Panoutsou, 2016)**

| | Mechanism | Marginal lands | Harvesting/ Collection | Logistics | Trade |
|---|---|---|---|---|---|
| **Regulatory** | Common Agricultural Policy | Early markets | Mature markets | | Mature markets |
| | Act on ecological products and farming practices | | Sustain markets | | Sustain markets |
| | Nitrate Directive (91/676/EEC) | | | | |
| | Certificate/Standardization | | Mature markets | | Mature markets |
| | | | Sustain markets | | Sustain markets |
| **Financial support** | Investment subsidies, direct payments | Early markets | Early markets | Early markets | |
| | R&D Grants | Early markets | Early markets | | |
| | Tax exemptions | | | Mature markets | Mature markets |
| | | | | Sustain markets | Sustain markets |
| **Information provision** | Strategies/Action plans | Early markets | Early markets | Early markets | Early markets |
| | Capacity building | Early markets | Early markets | Early markets | |
| | | Mature markets | Mature markets | Mature markets | |





**Appendix A**

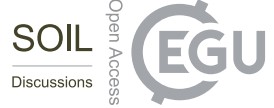

**Table A1: Properties of investigated soil profiles at the investigated case study sites.**

| | soil texture (FAO classes)[a] | bulk density[b] [g · cm⁻³] | rooting depth [cm] | profile available water [mm] | EC$_{SE}$[a] [mS · cm⁻¹] | pH$_{H2O}$[a] | Plant available nutrients[c] [mg · kg⁻¹] | | Heavy metal content[c] [mg · kg⁻¹] | | | | | |
|---|---|---|---|---|---|---|---|---|---|---|---|---|---|---|
| | | | | | | | P | K | As | Cr | Cu | Ni | Pb | Zn |
| GR Pel 1 | SCL/ SCL | 1.5 | 50 | 77.5 | 0.43/0.17 | 5.3/6.3 | 5 | 121 | 26 | 96 | 39 | 90 | - | 53 |
| GR Pel 2 | SL/SCL | 1.5 | 50 | 80.0 | 0.56/0.32 | 6.8/6.2 | 1 | 77 | 25 | 136 | 30 | 50 | - | 43 |
| GR Dro 1 | SCL/ CL | 1.5 | 40 | 60.0 | 0.64/0.18 | 6.4/5.7 | 4 | 175 | 107 | 123 | - | 60 | - | 50 |
| GR Dro 2 | SCL/ SCL | 1.3 | 30 | 54.0 | 0.28/0.16 | 6.0/5.7 | 4 | 172 | 113 | 125 | - | 49 | - | 41 |
| GR Sara | LS/LS | 1.7 | 60 | 102.0 | 1.40/0.43 | 5.3/5.7 | 4 | 44 | 23 | 320 | 36 | 215 | - | 49 |
| DE DB 1 | MS/MS | 1.8 | 80 | 56.0 | 2.44/1.44 | 8.0/7.2 | 381 | 59 | 6 | - | - | 37 | 38 | 52 |
| DE DB 2 | MS/MS | 1.8 | 80 | 56.0 | 1.91/0.56 | 7.3/5.8 | 53 | 24 | 7 | - | - | 35 | 56 | 42 |
| DE Wel 1 | SL/SL | 1.5 | 90 | 157.5 | 1.22/0.97 | 7.2/6.7 | 3 | 13 | 7 | - | - | - | 26 | 10 |
| DE Wel 2 | SL/LS | 1.8 | 70 | 105.0 | 0.71/1.16 | 6.0/4.0 | 5 | 36 | 4 | - | - | - | 24 | 13 |
| UA Pol 1 | HC/SC | 1.2 | 100 | 145.0 | 0.42/0.71 | 7.8/8.4 | 59 | 66 | 15 | 26 | - | - | 14 | 33 |
| UA Pol 2 | HC/HC | 1.6 | 140 | 182.0 | 0.37/0.11 | 6.1/6.4 | 93 | 114 | 8 | - | - | - | 25 | 23 |
| UA Vin 1 | HC/SC | 1.6 | 80 | 124.0 | 0.57/2.13 | 7.3/7.8 | 882 | 2574 | 19 | 46 | 20 | - | 27 | 70 |
| UA Vin 2 | C/SC | 1.7 | 80 | 124.0 | 0.49/3.29 | 7.5/8.0 | 891 | 416 | 20 | 64 | 36 | 40 | 113 | 132 |
| UA Vol A | LS/LS | 1.7 | 80 | 124.0 | 0.93/0.40 | 6.2/6.7 | 15 | 18 | 6 | - | - | - | 16 | 16 |
| UA Vol B | SCL/SCL | 1.7 | 80 | 88.0 | 0.78/0.54 | 7.7/7.9 | 19 | 44 | 5 | - | - | - | 23 | 9 |
| UA Vol C | LS/LS | 1.7 | 70 | 91.0 | 0.54/0.33 | 4.8/4.9 | 35 | 13 | 5 | - | 21 | - | 16 | - |
| UA Lvi A | CL/C | 1.9 | 30 | 42.0 | 1.40/0.79 | 7.5/7.9 | 4 | 36 | 9 | - | - | - | 23 | 17 |
| UA Lvi B | SL/SL | 1.8 | 90 | 166.5 | 1.43/1.53 | 7.8/7.9 | 23 | 83 | 9 | 17 | - | 35 | 15 | 41 |
| UA Lvi C | LS/MS | 1.7 | 60 | 57.0 | 1.59/0.34 | 6.5/5.7 | 23 | 8 | - | - | - | - | 22 | 7 |
| UA Lvi D | SCL/SCL | 1.8 | 90 | 130.5 | 0.89/0.77 | 7.5/7.8 | 6 | 22 | 5 | - | - | - | 21 | 27 |

data for: [a]topsoil/subsoil / [b]subsoil / [c]topsoil





**Table A2: Overview SQR hazard indicator (HI 1-11) multipliers and transformed influence values (*in italics*).**

| Site/ profile | HI 1 HI Multpl. | *Influence* | HI 2 HI Multpl. | *Influence* | HI 3 HI Multpl. | *Influence* | HI 4 HI Multpl. | *Influence* | HI 5 HI Multpl. | *Influence* | HI 6 HI Multpl. | *Influence* | HI 7 HI Multpl. | *Influence* | HI 8 HI Multpl. | *Influence* | HI 9 HI Multpl. | *Influence* | HI 10 HI Multpl. | *Influence* | HI 11 HI Multpl. | *Influence* |
|---|---|---|---|---|---|---|---|---|---|---|---|---|---|---|---|---|---|---|---|---|---|---|
| GR Pel 1 | 2.0 | *0.3* | 3.0 | *0.0* | 3.0 | *0.0* | 3.0 | *0.0* | 2.5 | *0.2* | 3.0 | *0.0* | 1.1 | *0.6* | 3.0 | *0.0* | 3.0 | *0.0* | 0.4 | *0.9* | 1.5 | *0.5* |
| GR Pel 2 | 2.0 | *0.3* | 3.0 | *0.0* | 3.0 | *0.0* | 3.0 | *0.0* | 2.0 | *0.3* | 3.0 | *0.0* | 1.1 | *0.6* | 3.0 | *0.0* | 3.0 | *0.0* | 0.7 | *0.8* | 1.5 | *0.5* |
| GR Dro 1 | 1.5 | *0.5* | 3.0 | *0.0* | 3.0 | *0.0* | 3.0 | *0.0* | 2.5 | *0.2* | 1.5 | *0.5* | 1.1 | *0.6* | 3.0 | *0.0* | 2.7 | *0.1* | 0.8 | *0.7* | 2.5 | *0.2* |
| GR Dro 2 | 1.5 | *0.5* | 3.0 | *0.0* | 3.0 | *0.0* | 3.0 | *0.0* | 2.5 | *0.2* | 0.5 | *0.8* | 1.1 | *0.6* | 3.0 | *0.0* | 3.0 | *0.0* | 0.5 | *0.8* | 1.9 | *0.4* |
| GR Sara 1 | 1.3 | *0.6* | 3.0 | *0.0* | 3.0 | *0.0* | 3.0 | *0.0* | 2.5 | *0.2* | 1.9 | *0.4* | 1.1 | *0.6* | 3.0 | *0.0* | 2.2 | *0.3* | 1.9 | *0.4* | 1.8 | *0.4* |
| DE DB 1 | 2.9 | *0.0* | 2.7 | *0.1* | 3.0 | *0.0* | 3.0 | *0.0* | 3.0 | *0.0* | 3.0 | *0.0* | 3.0 | *0.0* | 3.0 | *0.0* | 3.0 | *0.0* | 0.7 | *0.8* | 1.5 | *0.5* |
| DE DB 2 | 2.9 | *0.0* | 3.0 | *0.0* | 3.0 | *0.0* | 3.0 | *0.0* | 2.7 | *0.1* | 3.0 | *0.0* | 3.0 | *0.0* | 3.0 | *0.0* | 3.0 | *0.0* | 0.7 | *0.8* | 1.5 | *0.5* |
| DE Wel 1 | 3.0 | *0.0* | 1.0 | *0.7* | 3.0 | *0.0* | 2.1 | *0.3* | 1.9 | *0.4* | 3.0 | *0.0* | 3.0 | *0.0* | 3.0 | *0.0* | 3.0 | *0.0* | 3.0 | *0.0* | 3.0 | *0.0* |
| DE Wel 2 | 3.0 | *0.0* | 3.0 | *0.0* | 3.0 | *0.0* | 1.0 | *0.7* | 2.3 | *0.2* | 3.0 | *0.0* | 3.0 | *0.0* | 2.9 | *0.0* | 3.0 | *0.0* | 3.0 | *0.0* | 3.0 | *0.0* |
| UA Pol 1 | 3.0 | *0.0* | 3.0 | *0.0* | 2.5 | *0.2* | 3.0 | *0.0* | 3.0 | *0.0* | 3.0 | *0.0* | 3.0 | *0.0* | 3.0 | *0.0* | 3.0 | *0.0* | 3.0 | *0.0* | 3.0 | *0.0* |
| UA Pol 2 | 3.0 | *0.0* | 3.0 | *0.0* | 3.0 | *0.0* | 3.0 | *0.0* | 3.0 | *0.0* | 3.0 | *0.0* | 3.0 | *0.0* | 0.1 | *1.0* | 3.0 | *0.0* | 3.0 | *0.0* | 3.0 | *0.0* |
| UA Vin 1 | 2.8 | *0.1* | 2.9 | *0.0* | 3.0 | *0.0* | 3.0 | *0.0* | 3.0 | *0.0* | 3.0 | *0.0* | 2.8 | *0.1* | 2.0 | *0.3* | 3.0 | *0.0* | 2.5 | *0.2* | 2.5 | *0.2* |
| UA Vin 2 | 2.0 | *0.3* | 2.0 | *0.3* | 3.0 | *0.0* | 3.0 | *0.0* | 3.0 | *0.0* | 3.0 | *0.0* | 2.8 | *0.1* | 2.0 | *0.3* | 3.0 | *0.0* | 2.5 | *0.2* | 2.5 | *0.2* |
| UA Vol A | 3.0 | *0.0* | 3.0 | *0.0* | 3.0 | *0.0* | 3.0 | *0.0* | 1.9 | *0.4* | 3.0 | *0.0* | 3.0 | *0.0* | 2.5 | *0.2* | 3.0 | *0.0* | 3.0 | *0.0* | 3.0 | *0.0* |
| UA Vol B | 3.0 | *0.0* | 3.0 | *0.0* | 3.0 | *0.0* | 3.0 | *0.0* | 2.9 | *0.0* | 3.0 | *0.0* | 3.0 | *0.0* | 2.0 | *0.3* | 3.0 | *0.0* | 3.0 | *0.0* | 3.0 | *0.0* |
| UA Vol C | 3.0 | *0.0* | 3.0 | *0.0* | 3.0 | *0.0* | 1.5 | *0.5* | 2.7 | *0.1* | 3.0 | *0.0* | 3.0 | *0.0* | 3.0 | *0.0* | 3.0 | *0.0* | 3.0 | *0.0* | 3.0 | *0.0* |
| UA Lvi A | 3.0 | *0.0* | 3.0 | *0.0* | 3.0 | *0.0* | 3.0 | *0.0* | 2.5 | *0.2* | 1.0 | *0.7* | 3.0 | *0.0* | 3.0 | *0.0* | 3.0 | *0.0* | 3.0 | *0.0* | 3.0 | *0.0* |
| UA Lvi B | 2.9 | *0.0* | 3.0 | *0.0* | 3.0 | *0.0* | 3.0 | *0.0* | 2.9 | *0.0* | 3.0 | *0.0* | 3.0 | *0.0* | 2.0 | *0.3* | 3.0 | *0.0* | 3.0 | *0.0* | 3.0 | *0.0* |
| UA Lvi C | 3.0 | *0.0* | 3.0 | *0.0* | 3.0 | *0.0* | 3.0 | *0.0* | 1.9 | *0.4* | 3.0 | *0.0* | 3.0 | *0.0* | 3.0 | *0.0* | 3.0 | *0.0* | 3.0 | *0.0* | 3.0 | *0.0* |
| UA Lvi D | 3.0 | *0.0* | 3.0 | *0.0* | 3.0 | *0.0* | 3.0 | *0.0* | 1.9 | *0.4* | 3.0 | *0.0* | 3.0 | *0.0* | 2.0 | *0.3* | 3.0 | *0.0* | 3.0 | *0.0* | 3.0 | *0.0* |