# Peer review of "Assessment and quantification of marginal lands for biomass production in Europe using soil quality indicators"

_SOIL, 2018_

## Referee Comment (RC1) · Anonymous Referee #1 · 20 Jul 2018

General comments: The paper presents a methodology to assess and map marginal lands for biomass production for energy purposes using Muencheberg soil quality indicators (SQR) framework and GIS procedures. The demonstration of practical applicability of bioenergy production on marginal areas is carried on six European case study sites representing different types of marginal lands, method of bioenergy production and climate regimes. Moreover, SQR method is adapted for GIS analysis by means of pre-existent European database. As such, marginal lands potentials available for biomass production and for certain bioenergy crops have been mapped and quantified across Europe. The objective of the study is of outstanding interest not only for the scientific community but also for supporting European policies to identify areas

where incentives for expanding production of renewable resources without conflicting with agriculture production for human food and livestock feed. Moreover, the paper contains good use of English, very analytical description of the methods and results, and also discussion is exhaustive. According to my knowledge, the manuscript covers a topic that is relevant for the readership of SOIL journal and I recommend the manuscript accepted for publication with just minor changes and few integrations.

Specific comments: - Section 2.2.1, page 7, paragraph 6-17: I would suggest to shift these paragraphs to the discussion section. - Section 2.2.2, page 7, paragraph 25-30. I would suggest to describe why you choose 500m x 500m spatial resolution and the procedures adopted for downscaling/upscaling. Moreover, a reference to EPSG system should be provided. - Section 3.2.1, page 10, paragraph 1-5: You are encouraged to include some references on your assumption "these areas are, therefore, primarily not within the focus of the SQR assessment method". - Section 3.3, page 12, paragraph 20-24: when you state "the most frequent hazard indicator" you mean "the most extensive/widespread hazard indicator". Please, explain. - Section 3.3, page 13, paragraph 5-10. How you produced map in Figure 12? I guess some species/group of species might have overlapping growing conditions, resulting in overlaps of marginal lands suitability of these crop. Could you better explain how you dealt this issue?

---

## Author Comment (AC1) · 27 Aug 2018

Dear Referee #1,

we would like to thank you very much for your comments on our manuscript, particularly for your generally positive assessment of this contribution. Please find below our response to your specific remarks:

- Section 2.2.1, page 7, paragraph 6-17: I would suggest to shift these paragraphs to the discussion section.

=> We agree, these two paragraphs will be shifted to the end of chapter 3.3 in the

discussion section (p. 13).

- Section 2.2.2, page 7, paragraph 25-30. I would suggest to describe why you choose 500m x 500m spatial resolution and the procedures adopted for downscaling/upscaling. Moreover, a reference to EPSG system should be provided.

=> We agree. The paragraph in Section 2.2.2 after Table 4 will be modified as follows:

Pan-European datasets of the European Soil Data Center (ESDAC) have been primarily used whereas data from the HWSD were used for areas or parameters not covered by the ESDAC datasets, especially for Ukraine. The resolution of the original input datasets varied from 250 m to 5 km. A uniform cell size of 500m was applied to all datasets previous to the analysis. The resolution was selected following the resolution of the geospatial data available for soil texture classes from ESDAC. The selection was based on the fact that soil texture is itself one of the basic indicators for the calculation of SQR (B 1) and also a parameter for the calculation of two additional basic indicators (B 5 & B 6). Thus, the application of its resolution was selected to reflect substrate variations across Europe. Resampling for discrete data (e.g. land use) was performed using the nearest resampling algorithm whereas bilinear interpolation was applied for continuous data.

The coordinate reference system is ETRS89-LAEA Europe, EPSG:3035.

Latitude of Origin: 52 N Longitude of origin (Central Meridian): 10 E

Each raster dataset was reclassified based on the SQR field manual, the SQR Assessment scheme according to BGR (2010) and adaptations made by BTU CS within the SEEMLA project.

- Section 3.2.1, page 10, paragraph 1-5: You are encouraged to include some references on your assumption "these areas are, therefore, primarily not within the focus of the SQR assessment method".

=> The SQR method is originally restricted to assessing "soil's suitability for cropping

and grazing" (Mueller et al, 2007, p. 5) and is, therefore, focusing on cropland and grassland (Müller et al., 2010). For that reason the SQR indicators were chosen to validate the productivity function of soils and were, therefore, mainly applied to arable land (Henning et al., 2016). As wide parts of marginal lands, particularly those characterized by very low soil fertility, must be regarded as basically not suitable for traditional agriculture, their assessment was not within the primary focus of the SQR assessment method and methodological issues might arise. However, the SQR method can be generally applied to soils regardless of their quality, thus, also to soils of marginal lands and our study showed that the SQR system seems to be applicable also to assessing marginal lands. Against this background we suggest to modify the mentioned sentence in the manuscript as follows:

"The SQR system was primarily developed for valuating soil productivity functions related to traditional agricultural land use (Mueller et al., 2007, 2010) so that the assessment of land marginality is not within the original focus of the method."

- Section 3.3, page 12, paragraph 20-24: when you state "the most frequent hazard indicator" you mean "the most extensive/widespread hazard indicator". Please, explain.

=> In this paragraph the statistical analysis of the importance of the different hazard indicators is presented briefly. A more detailed description can be suggested as follows:

"Regarding marginal lands in Europe three SQR hazard indicators turned out to be most widespread (Tab. 7 and Fig. 11): 47.3 % of the marginal lands are characterized by shallow soils (H 6: soil depth above hard rock), 13.8 % are affected by unsuitable soil thermal regimes (H 12) and 3.2 % are endangered by drought risks (H 7). Shallow soils are frequent in the Mediterranean region as a result of extensive erosion processes in the past as well as in Scandinavia with young post-glacial soils. Drought risks are mainly restricted to the Iberian Peninsula whereas unsuitable soil thermal regimes are typical for the Northern parts of Scandinavia and the Alps, both with harsh climatic conditions."

- Section 3.3, page 13, paragraph 5-10. How you produced map in Figure 12? I guess some species/group of species might have overlapping growing conditions, resulting in overlaps of marginal lands suitability of these crop. Could you better explain how you dealt this issue?

=> The map shown in Fig. 12 is the result of applying the demands of selected bioenergy plant (as shown in Tab. 3) to the identified soil and site characteristics. Most signatures in this map indicate groups of potentially suitable bioenergy crops (e.g., basket willow is part of the upper three signatures of the legend, in each case combined with other crops). Thus, the map already shows several overlapping zones for some crops, e.g., willows and poplars could be cultivated alternatively (combined with different other crops) in wide parts of Europe. An additional sentence is suggested in the end of this paragraph to make this more obvious:

"Particularly, basket willows and poplars have large overlapping potential growing areas in Western and Central Europe and can be found, therefore, in different groups of bioenergy crops of Fig. 12."

References

Hennings, V., Höper, H., and Mueller, L.: Small-scale Soil Functional Mapping of Crop Yield Potentials in Germany, in: Mueller, L., Sheudshen, A.K., and Eulenstein, F. (Eds.): Novel Methods for Monitoring and Managing Land and Water Resources in Siberia, Springer, Germany, 597-617, 2016. Mueller, L., Schindler, U., Behrendt, A., Eulenstein, F., and Dannowski, R.: The Muencheberg Soil Quality Rating (SQR) Field Manual for detecting and assessing properties and limitations of soils for cropping and grazing, Müncheberg, Germany, 2007. Mueller, L., Schindler, U., Mirschel, W., Shepherd, T.G., Ball, B.C., Helming, K., Rogasik, J., Eulenstein, F., Wiggering, H.: Assessing the productivity function of soils. A review, Agron. Sustain. Dev., 30, 601-614, doi:10.1051/agro/2009057, 2010.

---

## Referee Comment (RC2) · Anonymous Referee #1 · 11 Sep 2018

Dear Editor, I've read the answers of the authors to my review. They provided exhaustive explanations on my request of explanations and proved their intent to make the suggested improvements to the paper. For all these reasons, I would recommend this manuscript to be accepted for publication.

---

## Referee Comment (RC3) · Anonymous Referee #2 · 24 Sep 2018

General comments: The manuscript presented focused on the Assessment and quantification of marginal lands for biomass production in Europe using soil quality indicators. The subject of this article is not novel, but the thematic is actual and interesting for publication. The research question is well explained, and it is relevant to the field of the journal. Moreover, the subject addressed in this article is worthy of investigation and provides new information and present a step forward in the knowledge. The contribution to the field, the technical quality, the quality of the presentation, the interest to readers and the added value of the current paper are good. But the depth of research of the paper needs some improvement. In my opinion, the results presented in the paper are important but they are merely described needing a wider discussion. In some

topics (namely section 3 and 4 of the results section) the discussion should be clearly improved, namely by doing comparisons with past studies and programs implemented (detailed explanation on how this can be done ahead of this revision). Claims are accurately supported by the results and they are reasonable. And in the conclusion, authors addressed to how this work can be continued. So, I recommend the manuscript to be published with the changes and revision made that should incorporate the suggestions and comments presented here.

Specific Comments: Detailed comments concerning the key elements of the paper and questions to be addressed are following below:

Abstract: It does reflect the content. It briefly presents the topic, state the scope of the experiment, and point out major findings.

Introduction: it provides a statement of the problem. It is clear what the authors hope to achieve, especially how this work may contribute to move the knowledge forward. However, the authors should put an emphasis in a sentence or two on how innovative is the methodology applied compared to previous studies and models.

Methodology: In order to replicate the research the information provided is clear. The design is suitable to answer the questions posed and was appropriate and the methodology was adequately described. However some things need clarification:

Section 2.1.3: page 5. Comments/questions: concerning soil contamination: only contamination with heavy metals was considered? Why not also contamination with hydrocarbons, pesticides, etc. Please provide an explanation in the manuscript to help readers with the same doubt. Page 5, lines 28.30, the sentence is confuse. Please rephrase. Page 7, lines 9-12, but those aspects are also included in this work? Perhaps it is better to indicate that those aspects are currently being studied. Just change the sentence to "that are currently being examined". So that readers understand that is work is still going on.

Results and Discussion/ Conclusion: Results are presented but the discussion needs improvement in some parts. Some examples on how this section can be improved: in section 3.3, authors should compare the results obtained with other studies that can show similarities or even contradictory aspects. This is important to show the importance of this study and how this study really represent an advancement to knowledge. Also in section 3, authors should also debate that not only correlation data between biomass yields and SQR scores are needed but also between biomass characteristics and SQR scores. Even when yields are high enough to be considered a feedstock, if the biomass does not have proper characteristics, processing it may be technically unfeasible. Authors should also debate that more correlations are needed with annual biomass crops. In this study only perennials were considered.

In section 4.2, it would be interesting to give some examples of success stories with other similar initiatives in the EU (concerning financial support to agricultural and agricultural related chains) and also the constraints and limitations derived from those initiatives, in order to show that that will be always pros and cons. In section 4, it should be also indicated that some of the regulations that are currently applied to biomass processes should be adapted to biomass processes that use biomass from marginal soils. Examples: targets on GHG emissions reduction.

---

## Author Comment (AC2) · 12 Oct 2018

Dear Referee #2,

We thank you very much for your suggested improvements of our manuscript. We will integrate your remarks into the final paper as follows:

Introduction: . . . However, the authors should put an emphasis in a sentence or two on how innovative is the methodology applied compared to previous studies and models.

=> With regard to the innovative aspect of our approach we would like to suggest adding the following additional paragraph after p. 3, l. 21:

[Figure]

"The SEEMLA approach presented in this paper is thought to contribute to the methodological development of assessment tools needed for step 1 (estimation of biomass technical potentials) of the analytical framework for evaluating sustainable biomass production potentials as proposed in a review by Batidzirai et al. (2012). Previous studies on global or European bioenergy potentials often tried to assess land availability for future bioenergy production mainly based on land use data and detectable changes derived by means of remote sensing methods (e.g., Campbell et al., 2008; Krasuska et al., 2010). However, Fritz et al. (2012) as well as Nalepa & Bauer (2012) demonstrated shortcomings of such approaches due to scaling problems. The here presented approach is based on an assessment of soil quality and related agricultural yield potentials using the SQR methodology. Results can reach a high spatial resolution depending on the availability and quality of input data. For Germany it had been demonstrated that high precision mapping of soil quality and related agricultural yield potentials is feasible (BGR, 2013). Additionally, this SEEMLA approach is supposed to allow for a clear differentiation between fertile agricultural lands and marginal lands with poor soil quality and weak agricultural yield potentials which are considered being still appropriate for bioenergy production. The SQR methodology explicitly includes numerous indicators for site related hazards for agricultural land use so that physico-chemical constraints of marginal lands and their severity can be directly revealed as demanded by Batidzirai et al. (2012)."

Methodology: . . . However some things need clarification: Section 2.1.3: page 5. Comments/questions: concerning soil contamination: only contamination with heavy metals was considered? Why not also contamination with hydrocarbons, pesticides, etc. Please provide an explanation in the manuscript to help readers with the same doubt.

=> Concerning contamination the SQR method refers for a first orientation to a methodology for sensory testing introduced by Lichtfuss (2004). This methodology provides several sensory parameters, e.g., soil colors or odor. Particularly, the latter can give clear indications of significant contamination with organic compounds (smell of petrol,

aromatic or phenolic compounds, etc.). In our case we did not find any suspicious odor within the investigated soil profiles during field assessment. Based on these findings it was concluded to concentrate on contamination with heavy metals which is not directly detectable by sensory testing. We suggest adding the following sentences to the manuscript:

p. 5, l. 24: "According to the SQR method hints for contaminations, particularly signs of artefacts, color or odor, can be tested roughly by means of sensory analysis (Lichtfuss, 2004). Suspicious colors or odor which could indicate possible contamination with organic compounds were not detectable within any of the investigated soil profiles, so that further analysis in the laboratory was restricted to possible inorganic contamination with heavy metals."

Page 5, lines 28.30, the sentence is confuse. Please rephrase.

=> The sentence will be changed as follows:

"Thus, regional project partners provided data on average biomass yields from adjacent field sites with soil conditions comparable to the respective case study sites. These data for the same bioenergy crops as cultivated on the case study sites were used as an estimate of local biomass yields."

Page 7, lines 9-12, but those aspects are also included in this work? Perhaps it is better to indicate that those aspects are currently being studied. Just change the sentence to "that are currently being examined". So that readers understand that is work is still going on.

=> The mentioned socioeconomic investigations are currently in progress and results are expected until the end of this year. We agree to the suggested modification of the sentence.

Results and Discussion/ Conclusion: Results are presented but the discussion needs improvement in some parts. Some examples on how this section can be improved: in

section 3.3, authors should compare the results obtained with other studies that can show similarities or even contradictory aspects. This is important to show the importance of this study and how this study really represent an advancement to knowledge.

=> We suggest adding the following text into l. 4 (p. 12):

"With this area size previous estimates of current land potentials for bioenergy production in Europe are clearly exceeded. Kluts et al. (2017) gave an overview on such studies. According to them , the minimum area of land currently available in Europe for bioenergy production was estimated as being clearly below 10 Mha. The maximum number was 30 Mha. The here presented approach only estimates the potential availability of land with poor or very poor soil quality which is considered not to be suitable for conventional agriculture. It must be assumed that an unknown proportion of this area is most probably also unsuitable for biomass production due to extreme site conditions. Thus, minimum soil quality for sustainable use of marginal lands has to be defined in future steps. For this purpose it will be necessary to further investigate the relationship between soil quality and biomass yield more precisely."

Also in section 3, authors should also debate that not only correlation data between biomass yields and SQR scores are needed but also between biomass characteristics and SQR scores. Even when yields are high enough to be considered a feedstock, if the biomass does not have proper characteristics, processing it may be technically unfeasible. Authors should also debate that more correlations are needed with annual biomass crops. In this study only perennials were considered.

=> This remark is important. In this project we did not investigate the quality of biomass produced at marginal lands. We also did not consider annual bioenergy crops. We suggest the following sentences to be added in section 3.2.1:

p. 10, l. 19: "The presented results are valid for perennial bioenergy crops, mainly for fast growing tree species. Effects of soil quality on the performance of annual bioenergy crops have not been considered. In addition, further research might be needed to

analyze relations between soil quality and characteristics of biomass with regard to its later use in power plants or bio refineries."

In section 4.2, it would be interesting to give some examples of success stories with other similar initiatives in the EU (concerning financial support to agricultural and agricultural related chains) and also the constraints and limitations derived from those initiatives, in order to show that that will be always pros and cons.

=> We suggest adding the following sentences at the end of section 4.2:

p. 14, l. 23: "Similar funding systems, e.g. the European Regional Development Fund (ERDF) may function as example. However, also in this case, aspects of a sustainable use of marginal lands with special focus on biomass production for bioenergy purposes need to be defined. In any case, it will be essential to bridge differences in agricultural and bioenergy policies in European countries, supporting underdeveloped regions, and avoiding an increase of land degradation by supporting a sustainable land management."

In section 4, it should be also indicated that some of the regulations that are currently applied to biomass processes should be adapted to biomass processes that use biomass from marginal soils. Examples: targets on GHG emissions reduction.

=> With regard to this remark we suggest modifying the last two sentences of section 4 (this has to be re-numbered to "5") as follows [additions are shown in brackets]:

Integrating bioenergy production at suitable marginal lands into future European policies (CAP) and the creation of suitable incentive programs might contribute to the objective to reach national and European renewable energy goals [for 2050] and to mitigate the rising land use conflict between the production of food and feed on the one hand and biomass on the other hand. It can be expected that the importance of marginal lands will increase during the next few decades as bioenergy is thought to play an important role for future energy supply in Europe [in terms of being able to

meet the targets on GHG emissions reduction until 2050].

References: Batidzirai, B., Smeets, E.M.W. and Faaij, A.P.C.: Harmonising bioenergy resource potentials – Methodological lessons from review of the state of the art bioenergy potential assessments, Renew. Sust. Energ. Rev., 16, 6598-6630, doi: 10.1016/j.rser.2012.09.002, 2012. BGR, Bundesanstalt für Geowissenschaften und Rohstoffe (Ed.): Map of the agricultural yield potential of German soils, Hannover, Germany, 2013. Campbell, E.J., Lobell, D.B., Genova, R.C. and Field, C.B.: The Global Potential of Bioenergy on Abandoned Agricultural Lands. Environ. Sci. Technol., 42, 5791-5794, doi:10.1021/es800052w, 2008,. Fritz, S., See, L., van der Velde, M., Nalepa, R.A., Perger, C., Schill, C., McCallum, I., Schepaschenko, D., Kraxner, F., Cai, X., Zhang, X., Ortner, S., Hazarika, R., Cipriani, A., Di Bella, C., Rabia, A.H., Garcia, A., Vakolyuk, M., Singha, K., Beget, M.E., Erasmi, S., Albrecht, F., Shaw, B. and Obersteiner, M.: Downgrading Recent Estimates of Land Available for Biofuel Production, Environ. Sci. Technol., 47, 1688-1694, doi:10.1021/es303141h, 2013. Kluts, I., Wicke, B., Leemans, R. and Faaij, A.: Sustainability constraints in determining European bioenergy potential: A review of existing studies and steps forward, Rew. Sust. Energ. Rev., 69, 719-734, doi:10.1016/j.rser.2016.11.036, 2017. Krasuska, E., Cadorniga, C., Tenorio, J., Testa, G., and Scordia, D.: Potential land availability for energy 5 crops production in Europe, Biofuel Bioprod. Bior., 4, 658–673, doi:10.1002/bbb.259, 2010. Lichtfuss, R.: Bodenkundlich-Sensorische Ansprache von Bodenproben, Ein Verfahren zur Beschreibung von Bodenproben, Bodenschutz, 1/04, 21-24, 2004. Nalepa, R.A. and Bauer, D.M.: Marginal lands: the role of remote sensing in constructing landscapes for agrofuel development, Jounal of Peasant Studies, 39, 403-422, doi: /10.1080/03066150.2012.665890, 2012.

---

## Author Comment (AC3) · 12 Oct 2018

We would like to thank you again for your valuable remarks and your positive reply to our suggested modifications of the manuscript.

---

## Author Response (AR1)

**Point-by-point response to the comments**

Reviewer remarks are marked in blue, author response and text revisions *[in italics]* are marked in red
(page and line numbers in reviewer comments and replies are according to the original manuscript)

[Rev. #2] Introduction: … However, the authors should put an emphasis in a sentence or two on how innovative is the methodology applied compared to previous studies and models.

5    => the following paragraph was added after p. 3, l. 21:

*The SEEMLA approach presented in this paper is thought to contribute to the methodological development of assessment tools needed for step 1 (estimation of biomass technical potentials) of the analytical framework for evaluating sustainable biomass production potentials as proposed in a review by Batidzirai et al. (2012). Previous studies on global or European bioenergy potentials often tried to assess land availability for future bioenergy production mainly based on land use data*

10    *and detectable changes derived by means of remote sensing methods (e.g., Campbell et al., 2008; Krasuska et al., 2010). However, Fritz et al. (2012) as well as Nalepa and Bauer (2012) demonstrated shortcomings of such approaches due to scaling problems. The here presented approach is based on an assessment of soil quality and related agricultural yield potentials using the SQR methodology. Results can reach a high spatial resolution depending on the availability and quality of input data. For Germany it had been demonstrated that high precision mapping of soil quality and related agricultural*

15    *yield potentials is feasible (BGR, 2013). Additionally, this SEEMLA approach is supposed to allow for a clear differentiation between fertile agricultural lands and marginal lands with poor soil quality and weak agricultural yield potentials which are considered being still appropriate for bioenergy production. The SQR methodology explicitly includes numerous indicators for site related hazards for agricultural land use so that physicochemical constraints of marginal lands and their severity can be directly revealed as demanded by Batidzirai et al. (2012).*

20    [Rev. #2] Section 2.1.3: page 5. Comments/questions: concerning soil contamination: only contamination with heavy metals was considered? Why not also contamination with hydrocarbons, pesticides, etc. Please provide an explanation in the manuscript to help readers with the same doubt.

=> Concerning contamination the SQR method refers for a first orientation to a methodology for sensory testing introduced by Lichtfuss (2004). This methodology provides several sensory parameters, e.g., soil colours or odour. Particularly, the

25    latter can give clear indications of significant contamination with organic compounds (smell of petrol, aromatic or phenolic compounds, etc.). In our case we did not find any suspicious odour within the investigated soil profiles during field assessment. Based on these findings it was concluded to concentrate on contamination with heavy metals which is not directly detectable by sensory testing. We suggest adding the following sentences to the manuscript:

he following sentences to the manuscript have been added (p. 5, l. 24):

*According to the SQR method hints for contaminations, particularly signs of artefacts colour or odour can be tested roughly by means of sensory analysis (Lichtfuss, 2004). Suspicious colours or odour which could indicate possible contamination with organic compounds were not detectable within any of the investigated soil profiles, so that further analysis in the laboratory was restricted to possible inorganic contamination with heavy metals.*

5 [Rev. #1] Page 5, lines 28.30, the sentence is confuse. Please rephrase.

=> The sentence has been changed as follows:

*Thus, regional project partners provided data on average biomass yields from adjacent field sites with soil conditions comparable to the respective case study sites. These data for the same bioenergy crops as cultivated on the case study sites were used as an estimate of local biomass yields.*

10 [Rev. #1] Section 2.2.1, page 7, paragraph 6-17: I would suggest to shift these paragraphs to the discussion section.

=> Both paragraphs have been shifted to the end of chapter 3.3 in the discussion section (p. 13).

[Rev. #2] Page 7, lines 9-12, but those aspects are also included in this work? Perhaps it is better to indicate that those aspects are currently being studied. Just change the sentence to "that are currently being examined". So that readers understand that is work is still going on.

15 => The sentence (now at the end of chapter 3.3) has been changed as recommended.

[Rev. #1] Section 2.2.2, page 7, paragraph 25-30. I would suggest to describe why you choose 500m x 500m spatial resolution and the procedures adopted for downscaling/upscaling. Moreover, a reference to EPSG system should be provided.

=> This paragraph has been modified as follows:

20 *Pan-European datasets of the European Soil Data Center (ESDAC) have been primarily used whereas data from the HWSD were used for areas or parameters not covered by the ESDAC datasets, especially for Ukraine. The resolution of the original input datasets varied from 250 m to 5 km. A uniform cell size of 500m was applied to all datasets previous to the analysis. The resolution was selected following the resolution of the geospatial data available for soil texture classes from ESDAC. The selection was based on the fact that soil texture is itself one of the basic indicators for the calculation of SQR (B 1) and*

25 *also a parameter for the calculation of two additional basic indicators (B 5 & B 6). Thus, the application of its resolution was selected to reflect substrate variations across Europe. Resampling for discrete data (e.g. land use) was performed using the nearest resampling algorithm whereas bilinear interpolation was applied for continuous data.*

*The coordinate reference system is ETRS89-LAEA Europe, EPSG:3035.*

*Latitude of Origin: 52 N    Longitude of origin (Central Meridian): 10 E*

30 *Each raster dataset was reclassified based on the SQR field manual, the SQR Assessment scheme according to BGR (2010) and adaptations made by BTU CS within the SEEMLA project.*

[Rev. #1] Section 3.2.1, page 10, paragraph 1-5: You are encouraged to include some references on your assumption "these areas are, therefore, primarily not within the focus of the SQR assessment method".

=> The SQR method is originally restricted to assessing "soil's suitability for cropping and grazing" (Mueller et al, 2007, p. 5) and is, therefore, focusing on cropland and grassland (Müller et al., 2010). For that reason the SQR indicators were chosen to validate the productivity function of soils and were, therefore, mainly applied to arable land (Henning et al., 2016). As wide parts of marginal lands, particularly those characterized by very low soil fertility, must be regarded as basically not suitable for traditional agriculture, their assessment was not within the primary focus of the SQR assessment method and methodological issues might arise. However, the SQR method can be generally applied to soils regardless of their quality, thus, also to soils of marginal lands and our study showed that the SQR system seems to be applicable also to assessing marginal lands. Against this background we have modified the mentioned sentence in the manuscript as follows:

*The basic aim of the SQR system is valuating soil productivity functions related to traditional agricultural land use (Mueller et al., 2007, 2010) so that the assessment of land marginality is not within the original focus of the method.*

[Rev. #2] Also in section 3, authors should also debate that not only correlation data between biomass yields and SQR scores are needed but also between biomass characteristics and SQR scores. Even when yields are high enough to be considered a feedstock, if the biomass does not have proper characteristics, processing it may be technically unfeasible. Authors should also debate that more correlations are needed with annual biomass crops. In this study only perennials were considered.

=> In this project we did not investigate the quality of biomass produced at marginal lands. We also did not consider annual bioenergy crops. We have added therefore the following sentences to section 3.2.1 (p. 10, l. 19):

*The presented results are valid for perennial bioenergy crops, mainly for fast growing tree species. Effects of soil quality on the performance of annual bioenergy crops have not been considered. In addition, further research might be needed to analyse relations between soil quality and characteristics of biomass with regard to its later use in power plants or bio refineries.*

[Rev. #2] in section 3.3, authors should compare the results obtained with other studies that can show similarities or even contradictory aspects. This is important to show the importance of this study and how this study really represent an advancement to knowledge.

*=> The following text has been added into l. 4 (p. 12):*

*With this area size previous estimates of current land potentials for bioenergy production in Europe are clearly exceeded. Kluts et al. (2017) gave an overview on such studies. According to them, the minimum area of land currently available in Europe for bioenergy production was estimated as being clearly below 10 Mha. The maximum number was 30 Mha. The here presented approach only estimates the potential availability of land with poor or very poor soil quality which is considered not to be suitable for conventional agriculture. It must be assumed that an unknown proportion of this area is most probably also unsuitable for biomass production due to extreme site conditions. Thus, minimum soil quality for*

*sustainable use of marginal lands has to be defined in future steps. For this purpose it will be necessary to further investigate the relationship between soil quality and biomass yield more precisely.*

[Rev. #1] Section 3.3, page 12, paragraph 20-24: when you state "the most frequent hazard indicator" you mean "the most extensive/widespread hazard indicator". Please, explain.

=> In this paragraph the statistical analysis of the importance of the different hazard indicators is presented briefly. The short original paragraph has been modified as follows (after line 14 of p. 12):

*Regarding marginal lands in Europe three SQR hazard indicators turned out to be most widespread (Tab. 7 and Fig. 11): 47.3 % of the marginal lands are characterized by shallow soils (H 6: soil depth above hard rock), 13.8 % are affected by unsuitable soil thermal regimes (H 12) and 3.2 % are endangered by drought risks (H 7). Shallow soils are frequent in the Mediterranean region as a result of extensive erosion processes in the past since antiquity as well as in Scandinavia with young post-glacial soils. Drought risks are mainly restricted to the Iberian Peninsula whereas unsuitable soil thermal regimes are typical for the Northern parts of Scandinavia and the Alps, both with harsh climatic conditions.*

[Rev. #1] Section 3.3, page 13, paragraph 5-10. How you produced map in Figure 12? I guess some species/group of species might have overlapping growing conditions, resulting in overlaps of marginal lands suitability of these crop. Could you better explain how you dealt this issue?

=> The map shown in Fig. 12 is the result of applying the demands of selected bioenergy plant (as shown in Tab. 3) to the identified soil and site characteristics. Most signatures in this map indicate groups of potentially suitable bioenergy crops (e.g., basket willow is part of the upper three signatures of the legend, in each case combined with other crops). Thus, the map already shows several overlapping zones for some crops, e.g., willows and poplars could be cultivated alternatively (combined with different other crops) in wide parts of Europe. An additional sentence has been added in the end of this paragraph to make this more obvious:

*Particularly, basket willows and poplars have large overlapping potential growing areas in Western and Central Europe and can be found, therefore, in different groups of bioenergy crops of Fig. 12.*

[Rev. #2] In section 4.2, it would be interesting to give some examples of success stories with other similar initiatives in the EU (concerning financial support to agricultural and agricultural related chains) and also the constraints and limitations derived from those initiatives, in order to show that that will be always pros and cons.

=> The following sentences have been added at the end of section 4.2 (p. 14, l. 23):

*Similar funding systems, e.g. the European Regional Development Fund (ERDF) may function as example. However, also in this case, aspects of a sustainable use of marginal lands with special focus on biomass production for bioenergy purposes need to be defined. In any case, it will be essential to bridge differences in agricultural and bioenergy policies in European countries, supporting underdeveloped regions, and avoiding an increase of land degradation by supporting a sustainable land management.*

[Rev. #2] In section 4, it should be also indicated that some of the regulations that are currently applied to biomass processes should be adapted to biomass processes that use biomass from marginal soils. Examples: targets on GHG emissions reduction.

=> With regard to this remark the last two sentences of section 4 have been modified as follows [additions are shown in brackets]:

[revised manuscript text omitted]
. | HI 1 Influence | HI 2 HI Multpl. | HI 2 Influence | HI 3 HI Multpl. | HI 3 Influence | HI 4 HI Multpl. | HI 4 Influence | HI 5 HI Multpl. | HI 5 Influence | HI 6 HI Multpl. | HI 6 Influence | HI 7 HI Multpl. | HI 7 Influence | HI 8 HI Multpl. | HI 8 Influence | HI 9 HI Multpl. | HI 9 Influence | HI 10 HI Multpl. | HI 10 Influence | HI 11 HI Multpl. | HI 11 Influence |
|---|---|---|---|---|---|---|---|---|---|---|---|---|---|---|---|---|---|---|---|---|---|---|
| GR Pel 1 | 2.0 | *0.3* | 3.0 | *0.0* | 3.0 | *0.0* | 3.0 | *0.0* | 2.5 | *0.2* | 3.0 | *0.0* | 1.1 | *0.6* | 3.0 | *0.0* | 3.0 | *0.0* | 0.4 | *0.9* | 1.5 | *0.5* |
| GR Pel 2 | 2.0 | *0.3* | 3.0 | *0.0* | 3.0 | *0.0* | 3.0 | *0.0* | 2.0 | *0.3* | 3.0 | *0.0* | 1.1 | *0.6* | 3.0 | *0.0* | 3.0 | *0.0* | 0.7 | *0.8* | 1.5 | *0.5* |
| GR Dro 1 | 1.5 | *0.5* | 3.0 | *0.0* | 3.0 | *0.0* | 3.0 | *0.0* | 2.5 | *0.2* | 1.5 | *0.5* | 1.1 | *0.6* | 3.0 | *0.0* | 2.7 | *0.1* | 0.8 | *0.7* | 2.5 | *0.2* |
| GR Dro 2 | 1.5 | *0.5* | 3.0 | *0.0* | 3.0 | *0.0* | 3.0 | *0.0* | 2.5 | *0.2* | 0.5 | *0.8* | 1.1 | *0.6* | 3.0 | *0.0* | 3.0 | *0.0* | 0.5 | *0.8* | 1.9 | *0.4* |
| GR Sara 1 | 1.3 | *0.6* | 3.0 | *0.0* | 3.0 | *0.0* | 3.0 | *0.0* | 2.5 | *0.2* | 1.9 | *0.4* | 1.1 | *0.6* | 3.0 | *0.0* | 2.2 | *0.3* | 1.9 | *0.4* | 1.8 | *0.4* |
| DE DB 1 | 2.9 | *0.0* | 2.7 | *0.1* | 3.0 | *0.0* | 3.0 | *0.0* | 3.0 | *0.0* | 3.0 | *0.0* | 3.0 | *0.0* | 3.0 | *0.0* | 3.0 | *0.0* | 0.7 | *0.8* | 1.5 | *0.5* |
| DE DB 2 | 2.9 | *0.0* | 3.0 | *0.0* | 3.0 | *0.0* | 3.0 | *0.0* | 2.7 | *0.1* | 3.0 | *0.0* | 3.0 | *0.0* | 3.0 | *0.0* | 3.0 | *0.0* | 0.7 | *0.8* | 1.5 | *0.5* |
| DE Wel 1 | 3.0 | *0.0* | 1.0 | *0.7* | 3.0 | *0.0* | 2.1 | *0.3* | 1.9 | *0.4* | 3.0 | *0.0* | 3.0 | *0.0* | 3.0 | *0.0* | 3.0 | *0.0* | 3.0 | *0.0* | 3.0 | *0.0* |
| DE Wel 2 | 3.0 | *0.0* | 3.0 | *0.0* | 3.0 | *0.0* | 1.0 | *0.7* | 2.3 | *0.2* | 3.0 | *0.0* | 3.0 | *0.0* | 2.9 | *0.0* | 3.0 | *0.0* | 3.0 | *0.0* | 3.0 | *0.0* |
| UA Pol 1 | 3.0 | *0.0* | 3.0 | *0.0* | 2.5 | *0.2* | 3.0 | *0.0* | 3.0 | *0.0* | 3.0 | *0.0* | 3.0 | *0.0* | 3.0 | *0.0* | 3.0 | *0.0* | 3.0 | *0.0* | 3.0 | *0.0* |
| UA Pol 2 | 3.0 | *0.0* | 3.0 | *0.0* | 3.0 | *0.0* | 3.0 | *0.0* | 3.0 | *0.0* | 3.0 | *0.0* | 3.0 | *0.0* | 0.1 | *1.0* | 3.0 | *0.0* | 3.0 | *0.0* | 3.0 | *0.0* |
| UA Vin 1 | 2.8 | *0.1* | 2.9 | *0.0* | 3.0 | *0.0* | 3.0 | *0.0* | 3.0 | *0.0* | 3.0 | *0.0* | 2.8 | *0.1* | 2.0 | *0.3* | 3.0 | *0.0* | 2.5 | *0.2* | 2.5 | *0.2* |
| UA Vin 2 | 2.0 | *0.3* | 2.0 | *0.3* | 3.0 | *0.0* | 3.0 | *0.0* | 3.0 | *0.0* | 3.0 | *0.0* | 2.8 | *0.1* | 2.0 | *0.3* | 3.0 | *0.0* | 2.5 | *0.2* | 2.5 | *0.2* |
| UA Vol A | 3.0 | *0.0* | 3.0 | *0.0* | 3.0 | *0.0* | 3.0 | *0.0* | 1.9 | *0.4* | 3.0 | *0.0* | 3.0 | *0.0* | 2.5 | *0.2* | 3.0 | *0.0* | 3.0 | *0.0* | 3.0 | *0.0* |
| UA Vol B | 3.0 | *0.0* | 3.0 | *0.0* | 3.0 | *0.0* | 3.0 | *0.0* | 2.9 | *0.0* | 3.0 | *0.0* | 3.0 | *0.0* | 2.0 | *0.3* | 3.0 | *0.0* | 3.0 | *0.0* | 3.0 | *0.0* |
| UA Vol C | 3.0 | *0.0* | 3.0 | *0.0* | 3.0 | *0.0* | 1.5 | *0.5* | 2.7 | *0.1* | 3.0 | *0.0* | 3.0 | *0.0* | 3.0 | *0.0* | 3.0 | *0.0* | 3.0 | *0.0* | 3.0 | *0.0* |
| UA Lvi A | 3.0 | *0.0* | 3.0 | *0.0* | 3.0 | *0.0* | 3.0 | *0.0* | 2.5 | *0.2* | 1.0 | *0.7* | 3.0 | *0.0* | 3.0 | *0.0* | 3.0 | *0.0* | 3.0 | *0.0* | 3.0 | *0.0* |
| UA Lvi B | 2.9 | *0.0* | 3.0 | *0.0* | 3.0 | *0.0* | 3.0 | *0.0* | 2.9 | *0.0* | 3.0 | *0.0* | 3.0 | *0.0* | 2.0 | *0.3* | 3.0 | *0.0* | 3.0 | *0.0* | 3.0 | *0.0* |
| UA Lvi C | 3.0 | *0.0* | 3.0 | *0.0* | 3.0 | *0.0* | 3.0 | *0.0* | 1.9 | *0.4* | 3.0 | *0.0* | 3.0 | *0.0* | 3.0 | *0.0* | 3.0 | *0.0* | 3.0 | *0.0* | 3.0 | *0.0* |
| UA Lvi D | 3.0 | *0.0* | 3.0 | *0.0* | 3.0 | *0.0* | 3.0 | *0.0* | 1.9 | *0.4* | 3.0 | *0.0* | 3.0 | *0.0* | 2.0 | *0.3* | 3.0 | *0.0* | 3.0 | *0.0* | 3.0 | *0.0* |